# ITERATIVE MULTI-OBJECTIVE POLICY OPTIMIZATION FOR ANTIBODY SEQUENCE DESIGN

## ABSTRACT

Antibodies are among the most important medicines in use today, yet their development is constrained by costly and labor-intensive affinity maturation. Computational antibody design offers a scalable alternative, but faces two central challenges: the lack of reliable affinity labels and the need to balance binding affinity with structural fidelity (self-consistency of refolded sequences to the original backbone). In this work, we formulate antibody sequence design as a multi-objective optimization problem and develop an iterative policy optimization framework tailored to this setting. To approximate experimental binding affinity, we construct a surrogate reward by regressing wet-lab $\Delta\Delta G$ measurements against Rosetta-derived interface metrics, including shape complementarity, buried surface area, and interfacial hydrogen bonds. To preserve structural fidelity, we introduce self-consistency RMSD as a complementary objective. Our method performs iterative training with a regression loss derived from the KL-regularized policy optimization objective, enabling stable on-policy learning under expensive structural evaluations and progressively guiding the policy toward Pareto-efficient trade-offs between binding affinity and structural fidelity. Across diverse antigen targets, this approach yields antibody sequences that achieve improved binding affinity while maintaining structural consistency, advancing computational antibody design toward practical therapeutic application.

## 1 INTRODUCTION

Antibodies constitute one of the most important classes of therapeutics, but their successful development often depends on labor-intensive affinity maturation. In nature, antibodies defend organisms by binding pathogens such as viruses and bacteria with high affinity (Murphy & Weaver, 2016), while engineered therapeutic antibodies are designed to neutralize diverse molecular targets, recruit immune effectors, or deliver drug payloads (Chiu & Gilliland, 2016). Binding strength and specificity are fine-tuned through affinity maturation, driven in vivo by somatic hypermutation and clonal selection (Victora & Nussenzweig, 2022); in vitro this is achieved by targeted mutagenesis and selection during antibody engineering (Tabasinezhad et al., 2019; Chiu & Gilliland, 2016).

Despite its importance, affinity maturation has not been explicitly incorporated into traditional *de novo* antibody design pipelines. Existing approaches typically separate backbone generation from sequence design, addressing affinity only in downstream refinement steps (Joubbi et al., 2024). We present **AbMPO**, which mirrors the goal of affinity maturation by updating a structure-conditioned sequence design policy to favor high-affinity sequences through iterative policy optimization. At each round, the current policy serves as the reference policy from which candidate sequences are sampled, refolded, and evaluated with Rosetta-derived metrics (Rohl et al., 2004). These evaluations provide reward signals for progressively improving the policy. By using large-batch offline sampling to amortize the cost of structure-based evaluations, our framework enables generation of higher-affinity antibody sequences without relying on post hoc optimization modules.

Any policy optimization method relies on a reliable reward signal to guide optimization. In antibody design, however, accurate binding affinity labels are scarce. While Rosetta's $\Delta\Delta G$ is widely used to assess variant quality (Barlow et al., 2018), its underlying $\Delta G$ estimates are often unreliable. Therefore, we construct a surrogate reward by regressing experimentally measured $\Delta\Delta G$ values against changes in Rosetta-derived interface metrics, i.e. shape complementarity (SC) (Lawrence & Col-

man, 1993), interface solvent-accessible surface area ($\Delta$SASA), and the number of cross-interface hydrogen bonds (Hbonds), between mutant and wild-type complexes. The resulting weighted combination yields a biologically grounded and interpretable approximation to binding affinity, calibrated directly against available wet-lab data.

While the surrogate affinity reward provides a practical signal for policy optimization, optimizing it alone often produces backbone inconsistency between the original scaffold and the refolded structure. In such cases, the generated antibody no longer matches its intended binding pose (Kessel & Ben-Tal, 2018), compromising structural self-consistency. To address this limitation, we introduce self-consistency RMSD (scRMSD) as a complementary objective alongside binding affinity. A naïve weighted sum of the two objectives could be used, but it requires arbitrary weight selection and fails to capture the full spectrum of trade-offs between affinity and structural fidelity. Instead, we encode the relative weighting directly into the policy optimization framework, conditioning the model on different trade-off preferences during training. This approach allows a single unified policy to explore a continuum of solutions and supports efficient discovery of the Pareto frontier spanning both surrogate binding affinity and structural fidelity.

Our pipeline achieves state-of-the-art performance across multiple antigens: PD1, SARS-CoV-2 RBD, IL7, and INSR. We make four contributions in this work:

- We introduce a unified framework that integrates affinity maturation directly into the structure-conditioned antibody sequence design process through iterative policy optimization, enabling efficient and stable exploration of sequence space.
- We develop a biologically grounded surrogate reward for binding affinity by calibrating Rosetta-derived interface metrics (SC, $\Delta$SASA, Hbonds) against experimental $\Delta\Delta G$ measurements.
- We extend the optimization to a multi-objective setting by incorporating scRMSD, allowing a single policy to recover the Pareto front balancing binding affinity and structural fidelity.
- Experiments demonstrate that AbMPO substantially outperforms state-of-the-art baselines, producing antibodies with stronger binding affinity to the target antigen while preserving structural consistency. We also show that our proposed method exhibits a logarithmic scaling behavior.

Together, these advances establish an end-to-end framework for *in silico* structure-conditioned antibody sequence design (inverse folding) that is both interpretable and practically effective.

## 2 RELATED WORK

**Multi-Objective Antibody Design**  Antibody design is inherently multi-objective, involving factors such as specificity, binding affinity, and beyond (Ye et al., 2024). Most recent approaches formulate antibody design as a sequence–structure co-design task, where CDR sequences and structures of existing antibodies are masked, and a generative model is trained to recover them (Jin et al., 2021; Kong et al., 2022; Luo et al., 2022; Kong et al., 2023; Zhou et al., 2024; Team et al., 2025a). Another related line of work leverages protein language models (pLM) trained specifically on antibody sequences, such as AbLang (Tobias H. Olsen & Deane, 2022), AntiBERTa (Leem et al., 2022), IgLM (Shuai et al., 2021), and nanoBERT (Hadsund et al., 2024). These models are typically used to mutate sequences in an attempt to improve binding affinity or sequence rationality (Hie et al., 2024). Reinforcement learning has also been explored on pLMs to optimize properties (Lee et al., 2025). Nonetheless, pLMs cannot incorporate antigen context, and thus fail to make antigen-specific mutations, limiting their effectiveness.

**Policy Optimization for Biological Sequence Design**  Policy optimization (Schulman et al., 2015; 2017; Rafailov et al., 2023; Richemond et al., 2024; Team et al., 2025b; Guo et al., 2025) has been applied to optimize diverse properties for biological sequence design (Angermueller et al., 2019; Chen et al., 2023; Wang et al., 2024). In the antibody domain, AB-Gen (Xu et al., 2023) fine-tunes a transformer policy via proximal policy optimization to generate CDR-H3 sequences under multi-property constraints. Prior structure-conditioned sequence design methods fail to incorporating downstream functional objectives (Dauparas et al., 2022). Recent extensions introduce additional signals, such as residue-level structural rewards (Xue et al., 2025), direct preference optimization with structural-similarity and diversity rewards(Park et al., 2024), and online reinforcement learning

with stability and diversity regularization (Wang et al., 2025). However, these approaches are not tailored to antibody-specific objectives and typically rely on a single metric or combine multiple metrics through simple weighted sums.

Prior work (Peng et al., 2019) also uses advantage information to bias updates, however, our approach is fundamentally different in formulation and learning dynamics. In contrast to ours, AWR performs a single-step weighted maximum likelihood estimation update, where the advantage appears only as an exponential weight on log-likelihood under a KL-derived constraint. As a result, our method supports stable, fully offline value estimation under a fixed reference policy, whereas AWR relies on implicit KL constraints and fixed-weight scalarization with no mechanism for conditioning on reward trade-offs.

Traditional multi-objective methods (Eichfelder, 2008; Van Moffaert et al., 2013; Giagkiozis & Fleming, 2015) (e.g. Chebyshev scalarization (Giagkiozis & Fleming, 2015), $\varepsilon$-constraint (Mavrotas, 2009), and gradient surgery (Liu et al., 2021)) usually require a separate policy (or scalarization run) for each trade-off. In contrast, our adaptive weight-conditioning differs from them by training a single policy that is explicitly conditioned on the weight vector. By embedding the weights and feeding them into the network, the policy learns a smooth, continuous mapping from desired trade-offs to optimal behaviors, enabling generalization to unseen weight vectors and eliminating the need for repeated optimization sweeps as in prior works. This makes our method more sample-efficient, more expressive, and capable of representing a dense Pareto front with a single model.

**Our work** addresses antigen-specific antibody design in an inverse folding setting, where the antigen–antibody backbone is given and the task is to infer the corresponding antibody sequence. Unlike prior methods, to provide a biologically grounded reward, we fit experimental $\Delta\Delta G$ values using changes in multiple Rosetta-derived interface metrics. Building on this, we formulate antibody design as a multi-objective optimization problem, explicitly encoding the trade-off between binding affinity and sequence–structure self-consistency within the policy optimization framework. This enables antigen-specific generation of high-quality candidates and dynamic balancing of objectives.

## 3 METHOD

In this section, we introduce **AbMPO**, an iterative policy optimization framework for multi-objective antibody sequence design (Figure 1). We begin by formalizing the problem and presenting the necessary preliminaries in Section 3.1. Next, we describe our policy optimization objective and the iterative training strategy that enables scalable on-policy learning in Section 3.2. In Section 3.3, we detail the design of multi-objective reward signals based on Rosetta metrics and structural consistency measure. Finally, in Section 3.4, we present our model design that adaptively balances competing objectives by learning along the Pareto front.

### 3.1 PRELIMINARIES

**Problem Formulation** We cast antibody sequence design as structure-conditioned sequence optimization. Let $L$ denote the sequence length and let $\mathcal{A}$ be a discrete amino-acid alphabet (e.g., the 20 canonical residues). We assume access to a fixed antigen–antibody backbone $B \in \mathcal{B}$, where $\mathcal{B}$ denotes the space of antigen-antibody complex backbones (e.g., 3D structures from PDB or predictions). Let $S_{\mathcal{I}_{\text{FR}}} \in \mathcal{A}^{|\mathcal{I}_{\text{FR}}|}$ denote a framework sequence, and let $\mathcal{I}_{\text{CDR}} \in \{1, \ldots, L\}$ be the set of designable positions corresponding to complementarity-determining regions (CDRs) (Murphy & Weaver, 2016). We define the *context* $C := (B, S_{\mathcal{I}_{\text{FR}}}, \mathcal{I}_{\text{CDR}})$. For a given context $C$, the feasible design space comprises all sequences of length $L$ thata preserve the fixed framework residues:

$$\mathcal{D}(C) = \left\{ \hat{S} \in \mathcal{A}^L : \hat{S}_{\mathcal{I}_{\text{FR}}} = S_{\mathcal{I}_{\text{FR}}} \right\}, \quad \text{where } \mathcal{I}_{\text{FR}} = \{1, \ldots, L\} \backslash \mathcal{I}_{\text{CDR}}.$$

A stochastic policy $\pi_\theta(\cdot \mid C)$ defines a distribution supported on $\mathcal{D}(C)$. Samples $\hat{S} \sim \pi_\theta(\cdot \mid C)$ are evaluated by a (possibly multi-objective) reward function:

$$r : \mathcal{C} \times \mathcal{A}^L \to \mathbb{R}^m, \quad (C, \hat{S}) \mapsto r(C, \hat{S}).$$

The learning problem is formulated as

$$\pi^* \in \arg\max_{\pi_\theta} \mathbb{E}_{C \sim D} \, \mathbb{E}_{\hat{S} \sim \pi_\theta(\cdot \mid C)} \big[ r(C, \hat{S}) \big], \tag{1}$$

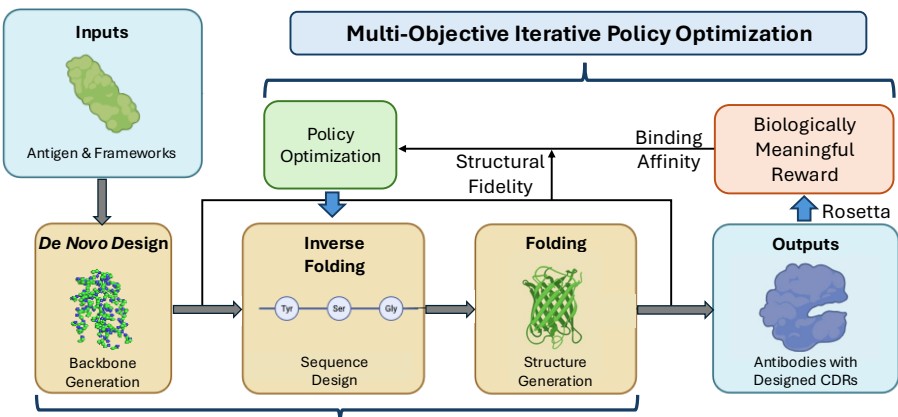

Figure 1: An overview of **AbMPO**. The pipeline begins with backbone generation of an antibody-antigen complex (*de novo* design), followed by structure-conditioned sequence design (inverse folding) and subsequent structure prediction (folding) to obtain the final antibody structure. The entire process is optimized using a multi-objective iterative policy optimization framework, guided by biologically meaningful rewards that approximate binding affinity and structural fidelity.

where $D$ is the distribution over contexts. Usually, if $r$ is vector-valued, a scalarization $g : \mathbb{R}^m \to \mathbb{R}$ is applied inside the expectation.

**KL-Regularized Policy Optimization**    To ensure stable training and prevent divergence from biologically plausible sequences, we regularize the policy $\pi_\theta$ (a neural policy parameterized by $\theta$) against a pre-trained reference distribution $\pi_{\text{ref}}$. The KL-regularized objective is defined as:

$$\max_{\pi_\theta} \; \mathbb{E}_{C \sim D, \hat{S} \sim \pi_\theta(\cdot|C)} \, r(C, \hat{S}) \; - \; \beta \, \text{KL}\big(\pi_\theta(\cdot|C) \, \| \, \pi_{\text{ref}}(\cdot|C)\big), \tag{2}$$

where $\beta > 0$ controls the strength of regularization. The optimal policy has the closed-form solution with corresponding optimal value function:

$$\pi^*(\hat{S} \,|\, C) \; \propto \; \pi_{\text{ref}}(\hat{S} \,|\, C) \, \exp\!\Big(\frac{r(C, \hat{S})}{\beta}\Big), \quad V^*(C) \; = \; \beta \ln \mathbb{E}_{\hat{S} \sim \pi_{\text{ref}}(\cdot|C)} \Big[\exp\!\Big(\frac{r(C, \hat{S})}{\beta}\Big)\Big]. \tag{3}$$

This formulation ensures that the learned policy maximizes the reward while remaining close to the pre-trained distribution. The proof is in Section A.

**Multi-Objective Optimization**    Multi-objective optimization (MOO) concerns problems in which multiple, often competing, objectives must be optimized simultaneously. Let the search space be $\mathcal{X} \subseteq \mathbb{R}^d$, where $d$ is the dimensionality of the design space, and let $f : \mathcal{X} \to \mathbb{R}^m$ denote a vector of $m$ objective functions. The goal is to identify solutions that achieve the best trade-offs among objectives. Formally, the problem is defined as

$$\text{Find } x^* \in \mathcal{X} \text{ such that no } x \in \mathcal{X} \text{ satisfies } f(x) \prec f(x^*), \tag{4}$$

where $\prec$ denotes Pareto dominance. A solution $x$ is said to *Pareto dominate* another solution $x^*$ if

$$\forall i \in \{1, \ldots, m\}, \; f_i(x) \leq f_i(x^*) \quad \text{and} \quad \exists j \in \{1, \ldots, m\} \text{ such that } f_j(x) < f_j(x^*). \tag{5}$$

In other words, $x$ is no worse than $x^*$ on all objectives and strictly better in at least one. A solution $x^*$ is *Pareto optimal* if it is not dominated by any other solution in $\mathcal{X}$. The set of all Pareto optimal solutions forms the *Pareto set (PS)*, and the corresponding set of objective vectors $\{f(x) \mid x \in PS\}$ is the *Pareto front (PF)*. The central goal of MOO is to approximate the PF as accurately as possible, thereby characterizing the spectrum of optimal trade-offs among competing objectives.

**Binding Affinity**    Binding affinity is a primary objective in antibody design, as it directly determines therapeutic efficacy and specificity (Nelson & Reichert, 2009). It is typically quantified by changes in binding free energy, $\Delta\Delta G$ (Gilson et al., 1997; Copeland, 2013). Here, $\Delta G$ denotes

the absolute binding free energy of an antibody–antigen complex, with more negative values corresponding to stronger interactions. In practice, affinity is often assessed through paired comparisons of a reference (often "wild-type") antibody sequence $S_{\mathrm{wt}}$ and a variant (mutant) sequence $S_{\mathrm{mut}}$. The relative change in binding is then defined as $\Delta\Delta G = \Delta G_{\mathrm{mut}} - \Delta G_{\mathrm{wt}}$ (Kortemme & Baker, 2002) where $\Delta G_{\mathrm{wt}}$ is the binding free energy of the wild type and $\Delta G_{\mathrm{mut}}$ is that of the variant. Positive values indicate reduced affinity (destabilization), while negative values indicate improved affinity (stabilization) (Guerois et al., 2002). Compared to $\Delta G$, which is sensitive to systematic errors from force fields, solvent models, or baseline calibration (Kellogg et al., 2011), $\Delta\Delta G$ is generally more robust because many such errors cancel when comparing closely related proteins. However, experimentally measured $\Delta\Delta G$ values are scarce and costly to obtain (Fowler & Fields, 2014), making them impractical as direct training signals. This motivates the construction of a surrogate reward that approximates $\Delta\Delta G$ using Rosetta-derived interface metrics calibrated against available experimental measurements (Kastritis & Bonvin, 2010).

### 3.2 ITERATIVE POLICY OPTIMIZATION FOR ANTIBODY SEQUENCE DESIGN

Antibody sequence design poses unique challenges for policy optimization: reward evaluation requires computationally intensive steps such as structure refolding and Rosetta scoring, and reliable affinity labels are scarce. To address these challenges, we develop an iterative policy optimization framework that is both sample-efficient and tailored to this domain. The algorithm is shown in Algorithm 1 and the detailed description is stated in Section E.

**Learning Objective** Building on the closed-form expression of the KL-regularized optimal policy and its associated value function (Equation (3)), we use the following identity, which links the reward advantage to the log-likelihood ratio between the optimal policy and the reference policy:

$$A(C, \hat{S}) := r(C, \hat{S}) - V^*(C) = \beta \log \frac{\pi^*(\hat{S} \mid C)}{\pi_{\mathrm{ref}}(\hat{S} \mid C)}. \tag{6}$$

This motivates fitting the current policy so that its log-likelihood ratio matches the observed advantage, leading to a regression loss. Concretely, given samples $\hat{S} \sim \pi_{\mathrm{ref}}(\cdot \mid C)$, we minimize the squared error

$$\mathcal{L}(\pi_\theta) = \mathbb{E}_{C \sim D} \, \mathbb{E}_{\hat{S} \sim \pi_{\mathrm{ref}}(\cdot \mid C)} \left[ \left( \beta \log \frac{\pi_\theta(\hat{S} \mid C)}{\pi_{\mathrm{ref}}(\hat{S} \mid C)} - \left( r(C, \hat{S}) - \hat{V}(C) \right) \right)^2 \right], \tag{7}$$

where $\hat{V}(C)$ is an empirical estimate of $V^*(C)$. This objective encourages $\pi_\theta$ to move toward a better advantage, while remaining close to $\pi_{\mathrm{ref}}$. Notably, both training data and value estimates are obtained **offline** from the reference policy. This offline design is particularly advantageous in antibody sequence generation, since large batches of sequences can be evaluated in parallel easily.

**Proposition 1.** *Let $\mathcal{L}(\cdot)$ be the regression loss defined in Equation (7). Assume that the value function used in Equation (7) is fixed and equal to $V^*$ (or a.s. equal on the support of $D$). Then $\pi^*$ is a global minimizer of $\mathcal{L}(\cdot)$. Moreover, let $p(\cdot \mid C)$ be any sampling distribution. If $\mathrm{supp}\, p(\cdot \mid C) = \mathrm{supp}\, \pi_{\mathrm{ref}}(\cdot \mid C)$ for all $C$ in the support of $D$, then $\pi^*$ is the unique global minimizer of $\mathcal{L}(\cdot)$.*

The proof is provided in Section B. This result formalizes the intuition behind Equation (7) and directly motivates the following remark.

**Remark** (Motivation). *With an exact value estimator ($\hat{V} = V^*$), the optimal solution of the KL-regularized problem (Equation (2)) also globally minimizes the regression objective (Equation (7)). In fact, the MSE term is zero under $\pi^*$, independent of the sampling distribution, showing that both objectives coincide at the true optimum.*

This relates the regression objective in Equation (7) to the KL-regularized policy objective in Equation (2). Although their gradients are not generally identical, differing in sign, scale, and the sampling distribution, they share the same closed-form optimal policy under an oracle baseline. In particular, when $\hat{V}(C)$ matches the soft value $V^*(C)$, both objectives induce the Boltzmann policy as the unique optimizer, and both population gradients vanish at that policy. This relation is formalized below; a full proof is provided in Section C.

**Proposition 2** (Shared Optimum but Dissimilar Gradients). *Let $\beta > 0$ and assume $\pi_{\text{ref}}(\cdot \mid C)$ has full support. Consider the KL-regularized objective in Equation (2) and the surrogate regression loss in Equation (7). With the oracle baseline $\hat{V}(C) = \beta \log \mathbb{E}_{\hat{S} \sim \pi_{\text{ref}}(\cdot|C)}[\exp(r(C, \hat{S})/\beta)]$, both objectives are uniquely optimized by the Boltzmann policy $\pi^*(\hat{S} \mid C) \propto \pi_{\text{ref}}(\hat{S} \mid C) \exp(r(C, \hat{S})/\beta)$, and both population gradients vanish at $\pi_\theta = \pi^*$. However, away from $\pi^*$ the two gradients are not generally proportional (differing in sign, scale, and sampling distribution), so they do not induce identical update directions or stationary sets in parameter space.*

**Iterative Updates**    To scale beyond a single update, we adopt an iterative scheme. At round $t$, the current policy $\pi_t$ is used to sample sequences $\{\hat{S}_i^t\}$, compute soft values $\hat{V}_t(C)$, and advantages $\hat{A}_t(C, \hat{S}) = r(C, \hat{S}) - \hat{V}_t(C)$. The next policy $\pi_{t+1}$ is then obtained by minimizing

$$\mathcal{L}_t(\pi_\theta) \;=\; \mathbb{E}_{C \sim D} \, \mathbb{E}_{\hat{S} \sim \pi_t(\cdot|C)} \left[ \left( \beta \log \frac{\pi_\theta(\hat{S} \mid C)}{\pi_t(\hat{S} \mid C)} - \hat{A}_t(C, \hat{S}) \right)^2 \right], \quad \text{for } t = 0, 1, \dots, T-1.$$

(8)

This '*resample–re-estimate–update*' loop gradually shifts probability mass toward high-advantage sequences while implicitly preserving close successive policies. For antibody sequence design, this iterative refinement is crucial: early iterations explore broadly from the reference distribution, while later iterations progressively adapt to capture subtle structural and binding improvements (see Section 4.2). The procedure thus combines the efficiency of offline evaluation with the adaptability of **on-policy learning**, making it especially well-suited for optimizing antibody design objectives.

**Remark** (On-policy learning and implicit trust region). *Although the regression loss contains no explicit KL penalty, minimizing $\mathcal{L}_t$ yields an update of the form $\pi_{t+1}(\cdot \mid C) \propto \pi_t(\cdot \mid C) \exp\{\hat{A}_t/\beta\}$. This update increases probability on high-advantage sequences while keeping $\pi_{t+1}$ close to its sampling distribution $\pi_t$. Because each round resamples from the current policy, the procedure resembles on-policy learning and inherits a built-in "trust region" effect controlled by $\beta$.*

### 3.3    Interpretable Surrogate Reward for Binding Affinity

To approximate experimentally measured $\Delta\Delta G$, we construct a surrogate reward based on *changes* in Rosetta-derived interface metrics between a mutant sequence and its wild type under the same context $C$. We focus on three metrics known to be predictive of binding affinity: SC, $\Delta$SASA, and Hbonds. These metrics capture complementary aspects of antibody–antigen binding: SC reflects overall geometric fit, $\Delta$SASA quantifies interface size and burial, and Hbonds capture specific directional interactions.

Formally, let $m_i(C, S) \in \{\text{SC}, \Delta\text{SASA}, \text{Hbonds}\}$ denote the $i$-th metric for sequence $S$ in context $C$, and let $S_{\text{wt}}$ be the wild-type sequence. After normalizing each metric to $[0, 1]$, we define

$$\Delta m_i(C, S) \;=\; m_i(C, S) - m_i(C, S_{\text{wt}}).$$

(9)

We then fit a linear regression model $(\mathbf{w}, b)$ to paired data $\{(C_j, S_{\text{mut}}^j, S_{\text{wt}}^j, \Delta\Delta G_j^{\text{exp}})\}$, yielding

$$\Delta\Delta G_j^{\text{exp}} \;\approx\; b + \sum_i w_i \, \Delta m_i(C_j, S_{\text{mut}}^j).$$

(10)

At inference, the surrogate reward for any candidate sequence $\hat{S}$ in context $C$ is computed as $R(\hat{S}, C) = b + \sum_i w_i \Delta m_i(C, \hat{S})$. Grounding the regression in experimental $\Delta\Delta G$ measurements calibrates the surrogate reward to real binding changes, while the linear formulation preserves interpretability by making each weight $w_i$ directly reflect the contribution of geometric complementarity, interface burial, and hydrogen bonding.

### 3.4    Multi-Objective Optimization: Binding Affinity vs. Structural Fidelity

A critical aspect of antibody design beyond binding affinity is structural self-consistency, measured by scRMSD between the refolded structure and the original backbone. Lower scRMSD indicates greater structural fidelity and stronger agreement with the intended backbone. Since scRMSD and

binding affinity capture complementary facets of design quality, both must be optimized jointly to ensure antibodies bind strongly while preserving structural fidelity.

To this end, we formulate the problem as an MOO problem, where the goal is to identify solutions that are Pareto optimal with respect to both binding affinity and structural fidelity. The resulting PF characterizes the spectrum of trade-offs, enabling the selection of designs that balance structural fidelity and binding strength rather than over-optimizing for a single metric.

**Fixed-Weight Scalarization**  A straightforward approach is to train separate policies with fixed scalarization weights. For each weight vector $\mathbf{p} = (p_1, p_2)$ with $p_1 + p_2 = 1$, the scalarized reward is defined as $r(\mathbf{p}, C, \hat{S}) = p_1 \cdot r_{\text{affinity}}(C, \hat{S}) + p_2 \cdot r_{\text{scRMSD}}(C, \hat{S})$, where $r_{\text{affinity}}$ is the normalized surrogate affinity reward and $r_{\text{scRMSD}}$ is the normalized negative scRMSD (used so that lower scRMSD contributes positively). A separate policy is trained for each $\mathbf{p}$, yielding discrete Pareto trade-off points, but limiting coverage and generalization to unseen preferences.

**Adaptive Weight-Conditioning**  To overcome the limitations of fixed-weight scalarization, here we propose a weight-conditioned approach. Instead of training multiple policies independently, we train a single policy conditioned on the weight vector $\mathbf{p}$, enabling adaptation to arbitrary trade-offs between binding affinity and structural fidelity. Specifically, we encode $\mathbf{p}$ into a smooth input embedding using an RBF kernel expansion:

$$\phi_k(\mathbf{p}) = \exp\left(-\frac{\|\mathbf{p} - \mu_k\|^2}{2\sigma^2}\right), \quad k = 1, \dots, K, \tag{11}$$

where $\{\mu_k\}_{k=1}^K$ are fixed centers placed on the weight simplex and $\sigma$ is a bandwidth parameter. The resulting embedding $\Phi(\mathbf{p}) = (\phi_1(\mathbf{p}), \dots, \phi_K(\mathbf{p}))$ is incorporated as an additional input to the policy network. Specifically, for MPNN, we assign $\Phi(\mathbf{p})$ as part of each node's feature representation, thereby conditioning the message-passing updates on the desired reward trade-off. Training then proceeds with the same advantage-matching loss as before, but now with $\mathbf{p}$ resampled at each update step, which can be formulated as follows:

$$\ell_p(\pi) = \mathbb{E}_{\mathbf{p} \sim \rho(\cdot)} \mathbb{E}_{C \sim D} \mathbb{E}_{\hat{S} \sim \pi_{\text{ref}}(\cdot | C)} \left[ \left( \beta \log \frac{\pi_t(\hat{S} \mid C, \mathbf{p})}{\pi_{\text{ref}}(\hat{S} \mid C)} - (\mathbf{p}^\top r(C, \hat{S}) - V^*(C)) \right)^2 \right], \tag{12}$$

where $r(C, \hat{S}) = (r_{\text{affinity}}(C, \hat{S}), r_{\text{scRMSD}}(C, \hat{S}))$. Because the policy now receives $\mathbf{p}$ as input, a single model can represent the entire trade-off surface. At inference time, we can sweep $\mathbf{p}$ across the simplex to approximate a dense set of Pareto-optimal solutions.

By conditioning the policy on a continuous trade-off vector $p$, one model can represent an entire continuum of operating points along the affinity–fidelity Pareto front. This yields several advantages: (1) a single universal policy that can be queried at inference with arbitrary preferences without retraining; (2) greater data and compute efficiency by sharing experience across all trade-offs; and (3) smooth interpolation between preferences, producing a dense and informative Pareto curve rather than isolated fixed-weight solutions. Our contribution is to bring this paradigm into structure-conditioned antibody design, showing that one preference-conditioned policy can replace many per-weight models without sacrificing biophysical performance.

# 4 EXPERIMENTS

## 4.1 EXPERIMENTAL SETUP

In this study, we focus on four representative antigen targets: PD1 *(Programmed Cell Death Protein 1)*, SARS-CoV-2 RBD *(Receptor Binding Domain of the SARS-CoV-2 Spike Protein)*, IL7 *(Interleukin-7)*, and INSR *(Insulin Receptor)*. For the overall design pipeline, we follow RFantibody (Bennett et al., 2024), leveraging antibody–antigen fine-tuned RFdiffusion (Watson et al., 2023) for backbone generation. For the structure-conditioned sequence design, which is the primary focus of this work, we adopt ProteinMPNN (Dauparas et al., 2022), the most widely used architecture for backbone-conditioned antibody sequence design.

Table 1: Overall performance on PD1 and SARS-CoV-2 RBD. Best results are shown in **bold**.

| Method | PD1 | | | | | | | SARS-CoV-2 RBD | | | | | | |
|---|---|---|---|---|---|---|---|---|---|---|---|---|---|---|
| | SC↑ | ΔSASA↑ | Hbonds↑ | $\Delta G_{sep}$↓ | scRMSD↓ | Sim↓ | Nat↑ | SC↑ | ΔSASA↑ | Hbonds↑ | $\Delta G_{sep}$↓ | scRMSD↓ | Sim↓ | Nat↑ |
| ESM-IF | 0.374±0.013 | 0.147±0.005 | 0.007±0.001 | 242.213±12.232 | 0.624±0.013 | 0.576±0.008 | 0.351±0.011 | 0.382±0.010 | 0.141±0.006 | 0.007±0.001 | 184.671±12.186 | 0.606±0.012 | 0.571±0.005 | 0.344±0.007 |
| AbLang | 0.405±0.013 | 0.153±0.006 | 0.009±0.001 | 228.371±15.893 | 0.581±0.014 | 0.534±0.005 | 0.337±0.011 | 0.413±0.013 | 0.147±0.006 | 0.008±0.001 | 174.425±19.572 | 0.564±0.010 | 0.529±0.005 | 0.354±0.007 |
| nanoBERT | 0.378±0.014 | 0.142±0.006 | 0.008±0.001 | 236.912±17.297 | 0.603±0.012 | 0.538±0.010 | 0.328±0.008 | 0.378±0.013 | 0.135±0.007 | 0.007±0.001 | 180.632±18.617 | 0.583±0.012 | 0.536±0.009 | 0.343±0.006 |
| InstructPLM | 0.427±0.011 | 0.151±0.006 | 0.008±0.001 | 205.219±12.328 | 0.572±0.009 | 0.543±0.006 | 0.364±0.010 | 0.436±0.012 | 0.146±0.006 | 0.009±0.001 | 152.314±12.670 | 0.539±0.015 | 0.545±0.008 | 0.358±0.010 |
| DiffAb | 0.408±0.011 | 0.159±0.006 | 0.008±0.001 | 225.191±13.795 | 0.567±0.015 | 0.545±0.008 | 0.312±0.009 | 0.423±0.013 | 0.158±0.007 | 0.008±0.001 | 168.149±12.290 | 0.527±0.008 | 0.551±0.007 | 0.319±0.008 |
| AbDPO | 0.412±0.011 | 0.164±0.007 | 0.010±0.001 | 178.582±13.147 | 0.557±0.013 | 0.541±0.007 | 0.319±0.011 | 0.427±0.012 | 0.163±0.006 | 0.010±0.001 | 132.857±15.614 | 0.518±0.010 | 0.542±0.008 | 0.331±0.011 |
| ProteinMPNN$_{AR}$ | 0.382±0.012 | 0.143±0.005 | 0.007±0.001 | 212.344±15.095 | 0.628±0.009 | 0.554±0.005 | 0.344±0.007 | 0.387±0.011 | 0.141±0.007 | 0.007±0.001 | 160.734±10.635 | 0.591±0.011 | 0.551±0.010 | 0.344±0.008 |
| ProteinMPNN$_{CMLM}$ | 0.395±0.005 | 0.149±0.007 | 0.008±0.001 | 198.763±17.207 | 0.604±0.013 | 0.546±0.008 | 0.338±0.008 | 0.399±0.013 | 0.147±0.005 | 0.008±0.001 | 152.582±14.537 | 0.589±0.015 | 0.539±0.009 | 0.346±0.008 |
| AbMPNN | 0.434±0.013 | 0.173±0.005 | 0.009±0.001 | 133.562±18.705 | 0.579±0.010 | **0.525±0.008** | 0.349±0.010 | 0.432±0.011 | 0.174±0.007 | 0.009±0.001 | 121.364±17.786 | 0.527±0.012 | 0.527±0.005 | 0.360±0.008 |
| **AbMPO** | **0.451±0.010** | **0.203±0.007** | **0.013±0.001** | **119.326±18.317** | **0.556±0.010** | 0.531±0.005 | **0.367±0.011** | **0.456±0.011** | **0.192±0.005** | **0.012±0.001** | **90.932±10.692** | **0.509±0.013** | **0.525±0.009** | **0.378±0.007** |
| *p-value* | 0.0491 | 5.25e$^{-5}$ | 0.00146 | 0.259 | 0.895 | 0.193 | 0.664 | 0.0483 | 0.00158 | 0.0133 | 0.0112 | 0.255 | 0.676 | 0.00534 |

**Dataset Curation**    We generated 256 backbones for each antigen and, during offline data sampling, decoded 16 sequences per backbone using the MPNN policy, resulting in 4,096 samples per antigen at each round. Rewards were computed with Rosetta (Rohl et al., 2004), and the resulting dataset was split into training and validation sets with a 90:10 ratio. The three Rosetta interface metrics used to construct the surrogate reward are described in detail in Section D.1.

**Training Details**    Our method is explicitly designed as a policy optimization fine-tuning on top of a strong supervised pre-training, analogous to how RLHF is applied on top of large language models after pre-training and supervised fine-tuning (SFT). Therefore, we initialize the policy from a SFT model trained on SAbDab (Dunbar et al., 2014) and OAS (Olsen et al., 2022), following the protocol of AbMPNN (Dreyer et al., 2023). The SFT initialization provides a strong and stable starting point by teaching the model to generate reasonable antibody sequences, thereby improving the efficiency and stability of subsequent policy optimization. We provide more information on hyper-parameters and other training details in Section D.2.

**Baselines**    We benchmark our approach against a broad set of baselines spanning both sequence-only and structure-conditioned methods. For pLM-based approaches, we include ESM-IF (Hsu et al., 2022), AbLang (Tobias H. Olsen & Deane, 2022), nanoBERT (Hadsund et al., 2024), and InstructPLM (Qiu et al., 2024), covering state-of-the-art pLMs with and without antibody-specific adaptation. For structure-conditioned design, we evaluate ProteinMPNN (Dauparas et al., 2022) in both autoregressive (AR) and conditional masked language modeling (CMLM) variants, AbMPNN (Dreyer et al., 2023), as well as structure-based methods including DiffAb (Luo et al., 2022) and AbDPO (Zhou et al., 2024). To ensure fair comparison with structure-based methods, sequences will be refolded using IgFold (Ruffolo et al., 2023). We provide more details in Section D.3.

**Evaluation Metrics**    We evaluate designs using both the optimization objectives and additional criteria. (1) For **binding affinity**, we report the three Rosetta interface metrics used in the surrogate reward—SC, ΔSASA, and *Hbonds*. Since these metrics approximate absolute binding free energy ($\Delta G$), we also report $\Delta G_{sep}$ computed by Rosetta. $\Delta G_{sep}$ quantifies the energy difference between the bound complex and the separated antibody–antigen structures, serving as a direct estimate of binding free energy. To reduce stochastic variability, all values are averaged over five Rosetta relaxations per design. (2) **Structural fidelity** is measured by *scRMSD*, which captures backbone agreement between the original scaffold and the refolded structure, with lower values indicating greater fidelity. (3) To evaluate antigen **specificity**, we compute sequence similarity across antibodies targeting different antigens. Lower similarity (*Sim*) reflects better specificity, as effective methods should produce distinct antibodies for distinct targets. (4) Finally, we assess **rationality** using inverse perplexity from AntiBERTy (Ruffolo et al., 2021), also referred to as naturalness (*Nat*), where higher values indicate closer alignment with natural antibody distributions. Unless otherwise noted, evaluations are reported on PD1 using 4,096 sampled designs. See more details in Section D.4.

Table 2: Overall performance on IL7 and INR across baselines. Best results are shown in **bold**.

| Method | IL7 | | | | | | | INR | | | | | | |
|---|---|---|---|---|---|---|---|---|---|---|---|---|---|---|
| | SC↑ | ΔSASA↑ | Hbonds↑ | $\Delta G_{sep}$↓ | scRMSD↓ | Sim↓ | Nat↑ | SC↑ | ΔSASA↑ | Hbonds↑ | $\Delta G_{sep}$↓ | scRMSD↓ | Sim↓ | Nat↑ |
| ESM-IF | 0.365±0.013 | 0.136±0.005 | 0.007±0.001 | 262.114±12.232 | 0.628±0.013 | 0.572±0.008 | 0.331±0.011 | 0.371±0.010 | 0.133±0.006 | 0.007±0.001 | 219.776±12.186 | 0.611±0.012 | 0.568±0.005 | 0.328±0.007 |
| AbLang | 0.394±0.013 | 0.147±0.006 | 0.009±0.001 | 247.331±15.893 | 0.592±0.014 | 0.530±0.005 | 0.342±0.011 | 0.401±0.013 | 0.143±0.006 | 0.009±0.001 | 198.155±19.572 | 0.576±0.010 | 0.536±0.005 | 0.347±0.007 |
| nanoBERT | 0.371±0.014 | 0.138±0.006 | 0.008±0.001 | 257.022±17.297 | 0.615±0.012 | 0.534±0.010 | 0.323±0.008 | 0.370±0.013 | 0.135±0.007 | 0.008±0.001 | 205.276±18.617 | 0.593±0.012 | 0.536±0.009 | 0.339±0.006 |
| InstructPLM | 0.419±0.011 | 0.146±0.006 | 0.008±0.001 | 224.532±12.328 | 0.582±0.009 | 0.538±0.006 | 0.357±0.010 | 0.423±0.012 | 0.144±0.006 | 0.009±0.001 | 174.832±12.670 | 0.549±0.015 | 0.542±0.008 | 0.354±0.010 |
| DiffAb | 0.400±0.011 | 0.153±0.006 | 0.008±0.001 | 245.129±13.795 | 0.577±0.015 | 0.540±0.008 | 0.307±0.009 | 0.415±0.013 | 0.152±0.007 | 0.008±0.001 | 192.251±12.290 | 0.537±0.008 | 0.547±0.007 | 0.315±0.008 |
| AbDPO | 0.404±0.011 | 0.158±0.007 | 0.010±0.001 | 204.789±13.147 | 0.568±0.013 | 0.536±0.007 | 0.314±0.011 | 0.419±0.012 | 0.159±0.006 | 0.008±0.001 | 182.893±15.614 | **0.526±0.010** | 0.538±0.008 | 0.327±0.011 |
| ProteinMPNN$_{AR}$ | 0.374±0.012 | 0.138±0.005 | 0.007±0.001 | 232.849±15.095 | 0.636±0.009 | 0.550±0.005 | 0.338±0.007 | 0.379±0.011 | 0.137±0.007 | 0.007±0.001 | 187.324±10.635 | 0.601±0.011 | 0.547±0.010 | 0.340±0.009 |
| ProteinMPNN$_{CMLM}$ | 0.387±0.005 | 0.143±0.007 | 0.008±0.001 | 217.908±17.207 | 0.612±0.013 | 0.543±0.008 | 0.332±0.008 | 0.392±0.013 | 0.142±0.005 | 0.008±0.001 | 178.215±14.537 | 0.597±0.015 | 0.535±0.009 | 0.343±0.008 |
| AbMPNN | 0.416±0.013 | 0.156±0.005 | 0.008±0.001 | 221.209±18.705 | 0.583±0.010 | 0.522±0.008 | 0.366±0.010 | 0.408±0.011 | 0.161±0.007 | 0.013±0.001 | 185.969±17.786 | 0.598±0.013 | 0.542±0.008 | 0.361±0.008 |
| **AbMPO** | **0.439±0.010** | **0.167±0.007** | **0.010±0.001** | **202.184±18.317** | **0.566±0.010** | **0.518±0.005** | **0.384±0.011** | **0.428±0.015** | **0.179±0.005** | **0.015±0.001** | **174.559±10.692** | 0.574±0.013 | **0.531±0.009** | **0.373±0.007** |
| *p-value* | 0.0169 | 0.0765 | 1.0000 | 0.803 | 0.792 | 0.371 | 0.0268 | 0.5766 | 0.00158 | 0.0133 | 0.972 | 0.000180 | 0.502 | 0.0356 |

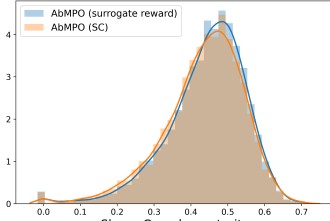 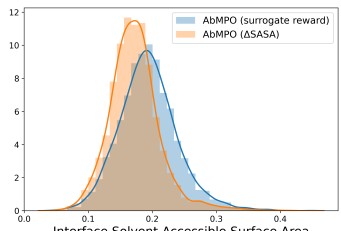 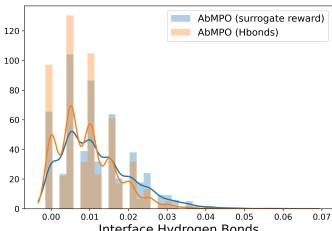

Figure 2: Optimization on the weighted surrogate reward leads to larger improvements in each individual metric (SC, $\Delta$SASA, and Hbonds) compared to optimizing them in isolation.

## 4.2 RESULTS AND ANALYSIS

**Validation of the Surrogate Reward** To validate our surrogate reward design, we first regress experimental $\Delta\Delta G$ values against four Rosetta-derived metrics: SC, $\Delta$SASA, Hbonds, and $\Delta G_{\text{sep}}$. The coefficient of $\Delta G_{\text{sep}}$ is close to zero and not statistically significant ($p$-value = 0.66), while variance inflation factors (VIF $\approx$ 1) indicate no collinearity among the variables. This suggests that SC, $\Delta$SASA, and Hbonds effectively capture the signal of $\Delta G$, which in turn approximates $\Delta\Delta G$. We therefore refit the regression using only these three metrics, yielding stable and interpretable coefficients: $-12.97$ for SC, $-38.26$ for $\Delta$SASA, and $-11.05$ for Hbonds. All coefficients are negative, indicating that higher values of these metrics correspond to stronger binding affinity (i.e., lower $\Delta\Delta G$). For clarity in constructing the weighted surrogate reward, we take the absolute values of these coefficients as weights. We provide full statistics in Section F.

We then evaluate the impact of the surrogate reward in preference optimization. As shown in Figure 2, optimizing on the weighted surrogate reward leads to substantial improvements across all three constituent metrics (SC, $\Delta$SASA, and Hbonds), surpassing the gains achieved when optimizing each metric individually. This demonstrates that the metrics are complementary and that their weighted combination provides a stronger and more informative training signal. Notably, improvements also transfer to $\Delta G_{\text{sep}}$, even though it is not directly included in the optimization objective, underscoring the close approximation of experimentally measured binding free energy using the surrogate reward (see Figure 3). Together, these results validate the surrogate reward as

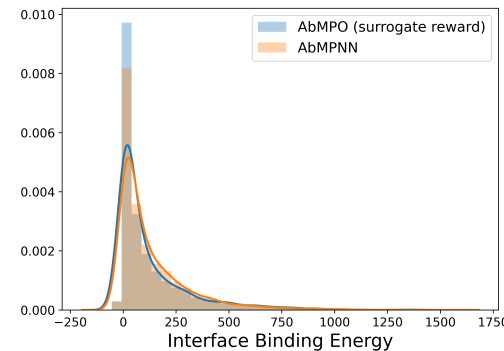

Figure 3: Improvements on $\Delta G_{\text{sep}}$ (not directly optimized) indicate that the surrogate reward closely approximates experimentally measured binding free energy.

both a biologically meaningful and computationally effective proxy for binding affinity, providing a reliable signal to guide antibody design.

**Benchmark Comparison** Across all four antigens (PD1, SARS-CoV-2 RBD, IL7, and INSR), AbMPO consistently achieves the best overall performance, ranking first in SC, $\Delta$SASA, Hbonds, and $\Delta G_{\text{sep}}$, which confirms that the surrogate reward effectively drives improvements in binding affinity. It also delivers the lowest Sim scores and the highest Nat scores, demonstrating its ability to generate antigen-specific, diverse, and natural sequences, surpassing even pLM baselines despite their large-scale pretraining. Among baselines, AbDPO stands out for strong results in scRMSD and competitive affinity-related metrics, reflecting the benefit of energy-aware preference optimization. AbMPNN also performs well on SC, $\Delta$SASA, Hbonds, and Nat, highlighting the advantage of antibody-specific adaptation in structure-conditioned models. Finally, InstructPLM achieves competitive results on SC and $\Delta G_{\text{sep}}$, showing that incorporating structural information into pLMs is important. We provide more analysis in Section F.1.

**Exploring the Pareto Front** Here we compare two proposed approaches to trace the affinity–fidelity trade-off. Figure 4a plots the PF over normalized $\Delta G_{\text{sep}}$ and scRMSD. The adaptive weight-conditioned policy, trained with sampled preference vectors, generates a dense cloud of

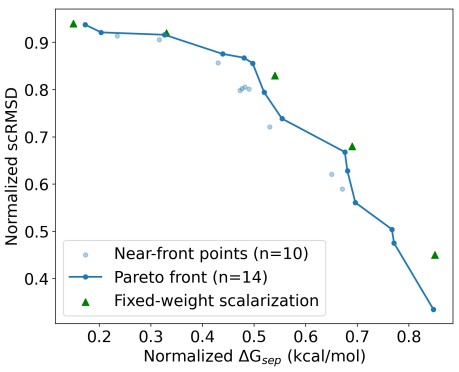
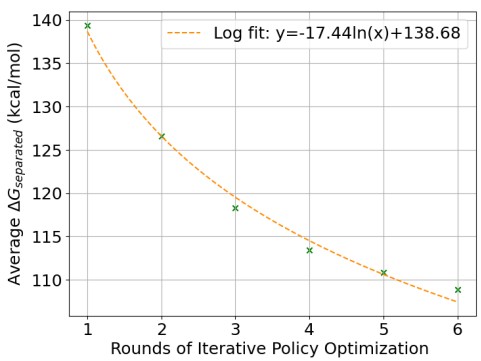

(a) Pareto front over normalized $\Delta G_{\text{sep}}$ and scRMSD comparing fixed-weight scalarization (green) and adaptive weight-conditioning (blue).

(b) Log-scaling of policy optimization: rapid early $\Delta G_{\text{sep}}$ gains followed by diminishing returns.

Figure 4: Comparison of Pareto front results and the scaling behavior of AbMPO.

near-front solutions and a smooth Pareto curve, yielding fine-grained control over trade-offs. The fixed-weighting scalarization produces a small number of discrete points; several of these lie on or slightly beyond the adaptive front, indicating stronger performance at those specific trade-offs. Conversely, some adaptive solutions are dominated and centered around [0.5 0.5], likely due to approximation/optimization noise and the difficulty of learning a universal mapping from preferences to policies. Overall, the comparison highlights a clear trade-off: weight-conditioning provides broad coverage and smooth interpolation along the front, whereas fixed-weight scalarization can deliver higher performance at specific operating points.

**Scaling Behavior** The average $\Delta G_{\text{sep}}$ decreases rapidly in early policy optimization rounds and slows thereafter, following a logarithmic scaling law that emerges when scaling up the number of optimization rounds. This behavior follows a favorable scaling law, where optimization becomes more efficient early on and steadily consolidates stronger binding designs in later rounds.

## 5 CONCLUSION AND DISCUSSION

We presented AbMPO, an iterative multi-objective policy optimization framework for structure-conditioned antibody sequence design. By grounding surrogate affinity rewards in experimental $\Delta\Delta G$ data and integrating structural fidelity through scRMSD, our method advances beyond single-objective or heuristic-weighted approaches. The iterative policy optimization algorithm enables explorative on-policy learning while maintaining efficiency, and the adaptive weight-conditioning mechanism supports fine-grained control over affinity–fidelity trade-offs. Across multiple antigens, AbMPO consistently outperforms existing baselines, producing sequences that are both high-affinity and structurally consistent. These results demonstrate the promise of policy optimization as a foundation for scalable, interpretable, and practical therapeutic antibody design.

AbMPO is applicable whenever we have (i) an antigen–antibody backbone, (ii) a structure-conditioned sequence model, and (iii) a structure-based evaluation oracle (Rosetta or similar). The algorithm is agnostic to the specific backbone generator and sequence model, and could in principle be combined with alternatives. However, the requirement for reliable structural evaluations and reasonable initial backbones is a genuine limitation.

Besides, when testing this approach in practice, for compact epitopes (PD1, RBD, IL7), the affinity and fidelity objectives are largely aligned; the policy can increase interface quality without leaving the designed backbone basin. For a more extended INSR epitope, these objectives conflict more strongly, and the policy without conditioning on pareto weights occasionally exploits backbone flexibility to gain affinity at the price of structural drift.

ETHICS STATEMENT

Our work focuses on computational antibody design, a field with significant potential for positive societal impact, including the development of life-saving therapeutics (e.g., antibodies against SARS-CoV-2) and antimicrobial peptides to address the global challenge of antibiotic resistance. At the same time, we acknowledge the dual-use nature of protein design research: in principle, the methods could be misapplied for harmful purposes, such as the engineering of toxic or pathogenic proteins. We believe the benefits of advancing safe and effective therapeutic discovery substantially outweigh these risks, and we stress that our research is intended for positive medical applications. We also recognize that the ultimate use of scientific knowledge cannot be fully controlled. With this awareness, we have carefully considered potential consequences and do not identify any immediate ethical concerns specific to the work presented here.

REPRODUCIBILITY STATEMENT

We provide a detailed description of all model architectures, training procedures, and hyperparameters in Section D to facilitate reproducibility. In addition, we plan to release our code upon acceptance of this paper, enabling the community to replicate our results and build upon our work.

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

# A  PROOF OF OPTIMAL POLICY AND VALUE FUNCTION

We consider the KL-regularized policy optimization objective:

$$\max_{\pi_\theta} \; \mathbb{E}_{C, \hat{S}\sim\pi_\theta(\cdot|C)}\Big[r(C,\hat{S})\Big] - \beta\,\mathrm{KL}\Big(\pi_\theta(\cdot|C)\,\|\,\pi_{\mathrm{ref}}(\cdot|C)\Big). \tag{13}$$

Since the expectation over $C$ simply averages across conditions, we can fix $C$ and study the per-condition problem:

$$\max_{\pi_\theta(\cdot|C)} \; \mathbb{E}_{\hat{S}\sim\pi_\theta(\cdot|C)}\Big[r(C,\hat{S})\Big] - \beta\,\mathrm{KL}\Big(\pi_\theta(\cdot|C)\,\|\,\pi_{\mathrm{ref}}(\cdot|C)\Big). \tag{14}$$

Expanding the KL term, $\mathrm{KL}\Big(\pi_\theta(\cdot|C)\,\|\,\pi_{\mathrm{ref}}(\cdot|C)\Big) = \sum_{\hat{S}} \pi_\theta(\hat{S}|C) \log \frac{\pi_\theta(\hat{S}|C)}{\pi_{\mathrm{ref}}(\hat{S}|C)}$, the objective becomes

$$J(\pi_\theta) = \sum_{\hat{S}} \pi_\theta(\hat{S}|C)\, r(C,\hat{S}) - \beta \sum_{\hat{S}} \pi_\theta(\hat{S}|C) \log \frac{\pi_\theta(\hat{S}|C)}{\pi_{\mathrm{ref}}(\hat{S}|C)}. \tag{15}$$

We impose the constraint $\sum_{\hat{S}} \pi_\theta(\hat{S}|C) = 1$ with multiplier $\lambda$, giving the Lagrangian

$$\mathcal{L}(\pi_\theta, \lambda) = \sum_{\hat{S}} \pi_\theta(\hat{S}|C)\, r(C,\hat{S}) - \beta \sum_{\hat{S}} \pi_\theta(\hat{S}|C) \log \frac{\pi_\theta(\hat{S}|C)}{\pi_{\mathrm{ref}}(\hat{S}|C)} + \lambda\Big(\sum_{\hat{S}} \pi_\theta(\hat{S}|C) - 1\Big).$$

Taking derivative with respect to $\pi_\theta$, we have $\frac{\partial\mathcal{L}}{\partial\pi_\theta(\hat{S}|C)} = r(C,\hat{S}) - \beta\Big(\log \frac{\pi_\theta(\hat{S}|C)}{\pi_{\mathrm{ref}}(\hat{S}|C)} + 1\Big) + \lambda$. Setting the above equation to zero, we can get $\log \frac{\pi_\theta(\hat{S}|C)}{\pi_{\mathrm{ref}}(\hat{S}|C)} = \frac{r(C,\hat{S})}{\beta} + \frac{\lambda-\beta}{\beta}$, and then exponentiating both sides we arrive at $\pi_\theta(\hat{S}|C) = \pi_{\mathrm{ref}}(\hat{S}|C)\, \exp\Big(\frac{r(C,\hat{S})}{\beta}\Big)\, \exp\Big(\frac{\lambda-\beta}{\beta}\Big)$.

The final exponential factor is a normalizing constant. Define the partition function $Z(C) = \sum_{\hat{S}} \pi_{\mathrm{ref}}(\hat{S}|C)\, \exp\Big(\frac{r(C,\hat{S})}{\beta}\Big)$, the **optimal policy** is

$$\pi^*(\hat{S}|C) = \frac{\pi_{\mathrm{ref}}(\hat{S}|C)\, \exp(r(C,\hat{S})/\beta)}{Z(C)} \;\; \propto \;\; \pi_{\mathrm{ref}}(\hat{S}|C) \exp(r(C,\hat{S})/\beta). \tag{16}$$

Following Equation (15), for any policy $\pi_\theta(\cdot|C)$,

$$J(\pi_\theta) = \beta \sum_{\hat{S}} \pi_\theta(\hat{S}|C) \left(\frac{r(C,\hat{S})}{\beta} + \log \pi_{\mathrm{ref}}(\hat{S}|C) - \log \pi_\theta(\hat{S}|C)\right) \tag{17}$$

$$= \beta\, \mathbb{E}_{\hat{S}\sim\pi_\theta(\cdot|C)}\left[\log \frac{\pi_{\mathrm{ref}}(\hat{S}|C)\, \exp(r(C,\hat{S})/\beta)}{\pi_\theta(\hat{S}|C)}\right]. \tag{18}$$

Given the (unnormalized) density

$$G_C(\hat{S}) \;:=\; \pi_{\mathrm{ref}}(\hat{S}|C)\, \exp\Big(r(C,\hat{S})/\beta\Big), \qquad Z(C) \;:=\; \sum_{\hat{S}} G_C(\hat{S}) = \sum_{\hat{S}} \pi_{\mathrm{ref}}(\hat{S}|C)\, e^{r(C,\hat{S})/\beta},$$

we can rewrite Equation (18) as follows

$$J(\pi_\theta) = \beta\, \mathbb{E}_{\hat{S}\sim\pi_\theta(\cdot|C)}\left[\log \frac{G_C(\hat{S})}{\pi_\theta(\hat{S}|C)}\right] \tag{19}$$

$$= \beta\, \mathbb{E}_{\hat{S}\sim\pi_\theta(\cdot|C)}\left[\log \frac{G_C(\hat{S})}{Z(C)} - \log \frac{\pi_\theta(\hat{S}|C)}{G_C(\hat{S})/Z(C)}\right] \tag{20}$$

$$= \beta\left(\log Z(C) - \mathrm{KL}\Big(\pi_\theta(\cdot|C)\,\Big\|\,\frac{G_C(\cdot)}{Z(C)}\Big)\right). \tag{21}$$

Since $\mathrm{KL}(\cdot \| \cdot) \geq 0$, we have $\mathbb{E}_{\hat{S} \sim \pi_\theta(\cdot|C)}\big[r(C, \hat{S})\big] - \beta \, \mathrm{KL}\big(\pi_\theta(\cdot|C) \,\|\, \pi_{\mathrm{ref}}(\cdot|C)\big) \;\leq\; \beta \log Z(C)$, with equality if and only if $\pi_\theta(\hat{S}|C) = \frac{G_C(\hat{S})}{Z(C)} = \frac{\pi_{\mathrm{ref}}(\hat{S}|C)\, e^{r(C,\hat{S})/\beta}}{\sum_{\hat{S}'} \pi_{\mathrm{ref}}(\hat{S}'|C)\, e^{r(C,\hat{S}')/\beta}}$. Therefore,

$$\max_{\pi_\theta(\cdot|C)} \left\{ \mathbb{E}_{\hat{S} \sim \pi_\theta(\cdot|C)}[r(C, \hat{S})] - \beta \, \mathrm{KL}\big(\pi_\theta(\cdot|C) \,\|\, \pi_{\mathrm{ref}}(\cdot|C)\big) \right\} = \beta \log Z(C). \tag{22}$$

Finally, note that $Z(C) = \sum_{\hat{S}} \pi_{\mathrm{ref}}(\hat{S}|C)\, e^{r(C,\hat{S})/\beta} = \mathbb{E}_{\hat{S} \sim \pi_{\mathrm{ref}}(\cdot|C)}\left[e^{r(C,\hat{S})/\beta}\right]$, the **optimal value** can be achieved by rewriting Equation (22):

$$V^*(C) = \beta \log \mathbb{E}_{\hat{S} \sim \pi_{\mathrm{ref}}(\cdot|C)}\left[\exp\left(\tfrac{r(C,\hat{S})}{\beta}\right)\right]. \tag{23}$$

**Remark.** We have an alternative derivation here. For any $\pi_\theta(\cdot|C)$,

$$\mathbb{E}_{\hat{S} \sim \pi_\theta}[r] - \beta \, \mathrm{KL}(\pi_\theta \| \pi_{\mathrm{ref}}) = \beta \, \mathbb{E}_{\hat{S} \sim \pi_\theta}\left[\log \frac{\pi_{\mathrm{ref}}(\hat{S}|C) e^{r/\beta}}{\pi_\theta(\hat{S}|C)}\right] \leq \beta \log \sum_{\hat{S}} \pi_{\mathrm{ref}}(\hat{S}|C) e^{r(C,\hat{S})/\beta},$$

by Jensen's inequality. Equality holds at $\pi_\theta = \pi^*$ in Equation (16).

## B  PROOF OF GLOBAL MINIMIZER OF THE SURROGATE REGRESSION LOSS

**Proposition.** *Let the surrogate regression loss be*

$$\mathcal{L}(\pi_\theta) := \mathbb{E}_{C \sim D} \, \mathbb{E}_{\hat{S} \sim p(\cdot|C)}\left[\big(r(C, \hat{S}) - V^*(C) - \beta \log \tfrac{\pi_\theta(\hat{S}|C)}{\pi_{\mathrm{ref}}(\hat{S}|C)}\big)^2\right], \tag{24}$$

*where $p(\cdot \mid C)$ satisfies $\mathrm{supp}\, p(\cdot \mid C) \subseteq \mathrm{supp}\, \pi_{\mathrm{ref}}(\cdot \mid C)$ and $\mathbb{E}_{\hat{S} \sim \pi_{\mathrm{ref}}(\cdot|C)}[\exp(r(C, \hat{S})/\beta)] < \infty$ for all $C$ in the support of $D$. Assume further that $\pi_\theta(\cdot \mid C) \ll \pi_{\mathrm{ref}}(\cdot \mid C)$ for all $C$. Then $\pi^*$ is a global minimizer of $\mathcal{L}(\pi_\theta)$. Moreover, if $\mathrm{supp}\, p(\cdot \mid C) = \mathrm{supp}\, \pi_{\mathrm{ref}}(\cdot \mid C)$ for all such $C$, then $\pi^*$ is the unique global minimizer.*

*Proof.* By Section 3.1, the KL-regularized optimal policy satisfies

$$r(C, \hat{S}) - V^*(C) = \beta \log \frac{\pi^*(\hat{S} \mid C)}{\pi_{\mathrm{ref}}(\hat{S} \mid C)}. \tag{25}$$

Substituting $\pi^*$ into Equation (24) makes the inner term zero pointwise $\pi_{\mathrm{ref}}(\cdot \mid C)$-almost surely, hence $\mathcal{L}(\pi^*) = 0$. Since $\mathcal{L}(\pi_\theta)$ is a sum of nonnegative terms, $\pi^*$ is a global minimizer.

For uniqueness, suppose $\tilde{\pi}_\theta$ also achieves $\mathcal{L}(\tilde{\pi}_\theta) = 0$. Then the squared term in Equation (24) must vanish $p(\cdot \mid C)$ for each $C$:

$$r(C, \hat{S}) - V^*(C) = \beta \log \frac{\tilde{\pi}_\theta(\hat{S} \mid C)}{\pi_{\mathrm{ref}}(\hat{S} \mid C)}. \tag{26}$$

If $\mathrm{supp}\, p(\cdot \mid C) = \mathrm{supp}\, \pi_{\mathrm{ref}}(\cdot \mid C)$, then Equation (26) holds $\pi_{\mathrm{ref}}$. Exponentiating yields

$$\tilde{\pi}_\theta(\hat{S} \mid C) \propto \pi_{\mathrm{ref}}(\hat{S} \mid C) \, \exp\big(r(C, \hat{S})/\beta\big).$$

Normalizing over $\hat{S}$ gives

$$\tilde{\pi}_\theta(\hat{S} \mid C) = \frac{\pi_{\mathrm{ref}}(\hat{S} \mid C) \exp\big(r(C, \hat{S})/\beta\big)}{\mathbb{E}_{\hat{S}' \sim \pi_{\mathrm{ref}}(\cdot|C)}\big[\exp(r(C, \hat{S}')/\beta)\big]} = \pi^*(\hat{S} \mid C).$$

for every $C$ in the support of $D$. Thus $\tilde{\pi}_\theta = \pi^*$ on the common support, proving uniqueness. $\square$

## C EQUIVALENCE BETWEEN OBJECTIVES AND THEIR GRADIENTS

**Proposition** (Shared optimum but non-proportional gradients in general). *Fix $\beta > 0$. Let $\pi_{\mathrm{ref}}(\cdot \mid C)$ have full support for each context $C$, and let $r(C, \hat{S})$ be bounded. For any baseline $\hat{V}(C)$ define the surrogate regression loss*

$$\mathcal{L}(\pi_\theta) = \mathbb{E}_{C \sim \mathcal{D}} \, \mathbb{E}_{\hat{S} \sim \pi_{\mathrm{ref}}(\cdot|C)} \left[ \left( \beta \log \frac{\pi_\theta(\hat{S}|C)}{\pi_{\mathrm{ref}}(\hat{S}|C)} - A(C, \hat{S}) \right)^2 \right], \quad A(C, \hat{S}) := r(C, \hat{S}) - \hat{V}(C). \tag{27}$$

*Consider also the KL-regularized policy objective*

$$\mathcal{J}(\pi_\theta) = \mathbb{E}_{C \sim \mathcal{D}} \left[ \mathbb{E}_{\hat{S} \sim \pi_\theta(\cdot|C)} \left[ r(C, \hat{S}) \right] - \beta \, \mathrm{KL}\big( \pi_\theta(\cdot \mid C) \, \| \, \pi_{\mathrm{ref}}(\cdot \mid C) \big) \right]. \tag{28}$$

*Assume: (i) $\pi_\theta(\hat{S} \mid C)$ is differentiable in $\theta$ with $\mathrm{supp}(\pi_\theta) \subseteq \mathrm{supp}(\pi_{\mathrm{ref}})$, (ii) $\hat{V}$ does not depend on $\theta$, and (iii) interchange of gradient and expectation is justified.*

*(A) Population gradients. The gradients can be written as*

$$\nabla_\theta \mathcal{L}(\pi_\theta) = 2\beta \, \mathbb{E}_{C \sim \mathcal{D}} \, \mathbb{E}_{\hat{S} \sim \pi_{\mathrm{ref}}(\cdot|C)} \left[ \left( \beta \log \frac{\pi_\theta}{\pi_{\mathrm{ref}}} - A \right) \nabla_\theta \log \pi_\theta(\hat{S} \mid C) \right], \tag{29}$$

$$\nabla_\theta \mathcal{J}(\pi_\theta) = \mathbb{E}_{C \sim \mathcal{D}} \, \mathbb{E}_{\hat{S} \sim \pi_\theta(\cdot|C)} \left[ \left( A - \beta \log \frac{\pi_\theta}{\pi_{\mathrm{ref}}} \right) \nabla_\theta \log \pi_\theta(\hat{S} \mid C) \right] \quad \text{when } \hat{V} \text{ is used as baseline.} \tag{30}$$

*In general, these two vector fields are* not *colinear because the sampling measures differ. In particular, after a change of measure,*

$$\nabla_\theta \mathcal{L}(\pi_\theta) = -2\beta \, \mathbb{E}_C \, \mathbb{E}_{\hat{S} \sim \pi_\theta(\cdot|C)} \left[ \underbrace{\frac{\pi_{\mathrm{ref}}(\hat{S}|C)}{\pi_\theta(\hat{S}|C)}}_{w(C, \hat{S})} \left( A - \beta \log \frac{\pi_\theta}{\pi_{\mathrm{ref}}} \right) \nabla_\theta \log \pi_\theta \right],$$

*which equals a weighted version of Equation (30) with importance weight $w$ and an overall sign/scale factor.*

*(B) Shared global maximizer under the oracle baseline. Let the* soft value

$$V^*(C) := \beta \log \mathbb{E}_{\hat{S} \sim \pi_{\mathrm{ref}}(\cdot|C)} \left[ \exp\big( r(C, \hat{S})/\beta \big) \right] \tag{31}$$

*and set $\hat{V} \equiv V^*$. Then the distributional optimization*

$$\max_{p(\cdot|C) \ll \pi_{\mathrm{ref}}(\cdot|C)} \mathbb{E}_{\hat{S} \sim p} [ r(C, \hat{S}) ] - \beta \, \mathrm{KL}\big( p \, \| \, \pi_{\mathrm{ref}}(\cdot \mid C) \big)$$

*has the unique solution*

$$\pi^*(\hat{S} \mid C) \, \propto \, \pi_{\mathrm{ref}}(\hat{S} \mid C) \, \exp\big( r(C, \hat{S})/\beta \big). \tag{32}$$

*Moreover, for this $\hat{V}$,*

$$\beta \log \frac{\pi^*}{\pi_{\mathrm{ref}}} = r - V^* \quad \Longrightarrow \quad \left( A - \beta \log \frac{\pi^*}{\pi_{\mathrm{ref}}} \right) \equiv 0 \quad \text{pointwise in } \hat{S},$$

*so both gradients Equation (29) and Equation (30) vanish pointwise, irrespective of the sampling measure. Hence, if the model class is realizable (i.e., there exists $\theta^*$ with $\pi_{\theta^*} = \pi^*$), then $\theta^*$ is a stationary point of both objectives.*

*(C) No general equality of stationary sets in parameter space. Away from $\pi^*$, the two population gradients differ by the non-constant importance weight $w$, so they are generally not scalar multiples of each other. Consequently, the sets of stationary parameters for $\mathcal{L}$ and $\mathcal{J}$ need not coincide, even though both share the same distributional maximizer $\pi^*$. The sign difference reflects descent on $\mathcal{L}$ versus ascent on $\mathcal{J}$ and is not merely cosmetic for stochastic optimizers.*

*Proof.* Under assumptions (i)–(iii), let

$$g_\theta(C, \hat{S}) := \beta \log \frac{\pi_\theta(\hat{S}|C)}{\pi_{\text{ref}}(\hat{S}|C)} - A(C, \hat{S}).$$

Then $\nabla_\theta g_\theta = \beta \nabla_\theta \log \pi_\theta(\hat{S} \mid C)$ and neither $A$ nor $\pi_{\text{ref}}$ depends on $\theta$. Differentiating the square in Equation (27) and taking expectations yields

$$\nabla_\theta \mathcal{L}(\pi_\theta) = 2 \, \mathbb{E}_{C, \hat{S} \sim \pi_{\text{ref}}} \left[ g_\theta(C, \hat{S}) \, \beta \, \nabla_\theta \log \pi_\theta(\hat{S} \mid C) \right],$$

which is Equation (29). For $\mathcal{J}$, apply the score-function identity with baseline $\hat{V}(C)$:

$$\nabla_\theta \mathcal{J}(\pi_\theta) = \mathbb{E}_{C, \hat{S} \sim \pi_\theta} \left[ \nabla_\theta \log \pi_\theta(\hat{S} \mid C) \left( A(C, \hat{S}) - \beta \log \frac{\pi_\theta(\hat{S}|C)}{\pi_{\text{ref}}(\hat{S}|C)} \right) \right],$$

giving Equation (30). Changing measure in Equation (29) from $\pi_{\text{ref}}$ to $\pi_\theta$ introduces $w = \frac{\pi_{\text{ref}}}{\pi_\theta}$ and the sign/scale, proving the non-colinearity claim in (A).

For part (B), fix a context $C$ and consider the distributional optimization

$$\max_{p(\cdot|C) \ll \pi_{\text{ref}}(\cdot|C)} \sum_{\hat{S}} p(\hat{S}) \, r(C, \hat{S}) - \beta \sum_{\hat{S}} p(\hat{S}) \log \frac{p(\hat{S})}{\pi_{\text{ref}}(\hat{S} \mid C)}.$$

Introducing a Lagrange multiplier for the constraint $\sum_{\hat{S}} p(\hat{S}) = 1$ yields the unique maximizer

$$\pi^*(\hat{S} \mid C) \propto \pi_{\text{ref}}(\hat{S} \mid C) \exp(r(C, \hat{S})/\beta),$$

i.e. the Boltzmann form Equation (32).

Now take $\hat{V}(C) = V^*(C)$, where $V^*(C)$ is given by Equation (31). A standard calculation shows that for this choice

$$\beta \log \frac{\pi^*(\hat{S} \mid C)}{\pi_{\text{ref}}(\hat{S} \mid C)} = r(C, \hat{S}) - V^*(C), \qquad \text{for all } \hat{S}.$$

Thus, in both gradients Equation (29) and Equation (30), the multiplicative factor

$$A(C, \hat{S}) - \beta \log \frac{\pi_\theta(\hat{S}|C)}{\pi_{\text{ref}}(\hat{S}|C)} = \left( r(C, \hat{S}) - V^*(C) \right) - \left( r(C, \hat{S}) - V^*(C) \right) = 0$$

vanishes *pointwise for every action* $\hat{S}$. This is stronger than merely vanishing in expectation: the integrand is identically zero, so both gradients are exactly zero regardless of whether samples are drawn from $\pi_{\text{ref}}$ or from $\pi_\theta$.

Consequently, whenever the model class can represent $\pi^*$, any parameter $\theta^*$ satisfying $\pi_{\theta^*} = \pi^*$ is a stationary point of both objectives.

Finally, note why this argument does *not* imply that the two gradients are proportional everywhere: for a general $\pi_\theta \neq \pi^*$, the regression gradient can be rewritten under $\pi_\theta$ only at the cost of introducing the non-constant importance weight $w(C, \hat{S}) = \frac{\pi_{\text{ref}}(\hat{S}|C)}{\pi_\theta(\hat{S}|C)}$. Hence the two expectations are not scalar multiples of each other, and a zero of one gradient need not be a zero of the other. This justifies statement (C).

□

# D    MORE DETAILS ON EXPERIMENTAL SETUP

## D.1    DATA CURATION SETUP

To curate training data for antibody design, we follow the four-stage pipeline (Bennett et al., 2024), ensuring structural and sequence consistency across rounds of optimization:

**Input Representation**    Antibody–target complexes are stored in an `HLT` file, which is a PDB file annotated with: (1) Heavy (`H`), Light (`L`), and Target (`T`) chain IDs. (2) Ordered chain arrangement: Heavy → Light → Target. (3) Explicit `REMARK` entries specifying the residue indices of each CDR loop. These annotations allow consistent information flow between modules. Chothia-annotated structures from SAbDab (Dunbar et al., 2014) can be converted into `HLT` format.

**Data Generation**    Each training round consists of four sequential steps to sample offline data and evaluate their rewards:

1. RFdiffusion: An antibody-finetuned diffusion model generates backbone structures against specified epitopes. CDRs are flexibly designed within user-defined length ranges, while non-loop framework regions remain fixed.

2. AbMPNN: Designed backbones are converted to sequences by assigning CDR residues. This ensures sequence variability while preserving framework integrity.

3. RoseTTAFold2 (RF2) Here we use an antibody-finetuned RF2 model for structure prediction (folding). Predicted structures are generated for each designed sequence, serving as a quality filter to assess whether designs fold as intended.

4. Rosetta Analysis: Refolded complexes are evaluated with Rosetta's `InterfaceAnalyzer`. Three interface metrics, shape complementarity (SC), interface solvent-accessible surface area ($\Delta$SASA), and the number of cross-interface hydrogen bonds (Hbonds), are extracted. These are later combined into weighted reward functions for optimization.

**RFdiffusion Inference Setup**    For antibody fine-tuning with RFdiffusion (Watson et al., 2023), we set the following arguments for generating antibody backbones:

- Antibody CDRs: `antibody.design_loops=[H1:6-10,H2:7-11,H3:5-20]`. We follow Bennett et al. (2024) to set the length of CDR-H1, H2, H3 in a variable range: CDR-H1 (length 6–10), H2 (length 7–11), and H3 (length 5–20).

- Diffusion Steps: `diffuser.T=50`. The total diffusion process will be a total of 50 steps.

- Inference Step: `inference.final_step=48`. The denoising trajectory is started at step 48, rather than running all 50 steps, which can speed up inference by a large margin.

- Deterministic Inference: `inference.deterministic=True`. Inference is performed deterministically, disabling stochastic noise, which also makes inference faster.

## D.2    TRAINING SETUP

For training, we adopted the following hyperparameter configuration. We set $\beta = 1/2$ (since this value ensures the nice property of gradient as stated in Theorem 2), and a temperature of $10^{-4}$. The learning rate was $5 \times 10^{-7}$ with a weight decay of $1 \times 10^{-5}$, and training ran for 200 epochs. We use a batch size of 16 sequences per backbone. To stabilize optimization, we applied gradient clipping with a loss clip value of 10. Validation was performed with a ratio of 0.1. The model parameters were initialized from a pretrained MPNN checkpoint following Dreyer et al. (2023) for both reference and online policies. We fixed the random seed during training to ensure the reproducibility of the results. For training and inference, we use 8 NVIDIA L40S GPUs, which have 384 GB GPU memory in total (48 GB per GPU).

For regularization and scheduling, we employed a cosine learning rate schedule with warmup. For the radial basis function (RBF) kernel expansion, we leveraged the 128-dimensional node features from the MPNN backbone by allocating 64 dimensions to each objective. We distributed 64 centers evenly in the range $[0, 1]$, with spacing denoted as $\delta$, and set the bandwidth parameter to $\sigma = \delta$. A lightweight hyperparameter sweep indicated that this configuration yielded the most robust performance across objectives.

## D.3    BASELINES

We benchmark against sequence-only methods (pLMs), structure-conditioned sequence design methods, and structure-based methods. Unless otherwise noted, we follow each method's public implementation with default settings, enforce the same total sampling budget per antigen, and evaluate all candidates with the same downstream scoring pipeline described in Section D.1. For fairness across methods, all sequences are refolded with IgFold (Ruffolo et al., 2023) before structure-based evaluation.

**Sequence-only methods (pLMs).**

1. **ESM-IF** (Hsu et al., 2022). ESM-IF is a structure-conditioned protein design model that predicts sequences from backbone coordinates. Built on large-scale protein language models, it accurately recovers native sequences and generalizes across diverse structural contexts, enabling scalable inverse folding for protein design.

2. **AbLang** (Tobias H. Olsen & Deane, 2022). AbLang is trained specifically on antibody sequences. We use the heavy-chain model to suggest token-level edits under a masked-LM objective while keeping framework constraints (e.g., conserved cysteines) intact. Promising mutations are identified over nine rounds, collecting all sequences that pass the LM scoring filter.

3. **nanoBERT** (Hadsund et al., 2024). Given our focus on single-domain antibodies (sdAbs), we include nanoBERT, a pLM pre-trained on sdAb sequences. We run nine rounds of mutation identification and collect candidates that pass the model's confidence filter each round.

4. **InstructPLM** (Qiu et al., 2024). The paper introduces InstructPLM, a protein design framework that aligns protein language models (pLMs) with structural information through cross-modality alignment and instruction fine-tuning. Instead of training full models from scratch, InstructPLM freezes a backbone encoder and a large pLM decoder, adding only a lightweight cross-attention adapter. This setup enables the model to follow structure-based prompts and generate variable-length protein sequences.

**Structure-conditioned sequence design methods.**

1. **ProteinMPNN** (AR & CMLM) (Dauparas et al., 2022). We evaluate both the standard autoregressive (AR) variant and the conditional masked language modeling (CMLM) variant. Both are conditioned on the antibody–antigen complex backbone; we design variable regions while fixing protected residues. We sample multiple candidates per complex using the authors' recommended decoding settings and select by ProteinMPNN internal score before downstream evaluation.

2. **AbMPNN** (Dreyer et al., 2023). An antibody-specific inverse folding model fine-tuned from ProteinMPNN to design antibody sequences from structural backbones. By training on both predicted structures from the Observed Antibody Space and experimentally resolved complexes from SAbDab, the model shows enhanced structural designability, stability of heavy–light chain interfaces, and better preservation of canonical loop conformations compared to ProteinMPNN.

**Structure-based methods.**

1. **DiffAb** (Luo et al., 2022). DiffAb trains a diffusion model over residue identities, coordinates, and orientations to optimize antibody sequences within the antibody–antigen complex. We follow the perturb-and-denoise procedure on the complex, generate ten designs per antigen, and use our predictor to select the top three for evaluation, matching the procedure used in the original description.

2. **AbDPO** (Zhou et al., 2024). Built on DiffAb, AbDPO fine-tunes the diffusion model with residue-level decomposed energy preferences via direct preference optimization to bias toward low-energy samples. Sampling and selection mirror DiffAb under the same candidate budget (ten per antigen, top three forwarded). We reproduce this method given the official codebase is not released.

### D.4 EVALUATION METRICS

We evaluate antibody designs using both the optimization objectives and a suite of additional quality criteria. To reduce stochastic variability, Rosetta-based metrics are averaged over five independent relaxations per design. Our evaluation framework captures four broad aspects of design quality: (1) **Binding affinity**, quantified through both surrogate interface metrics and direct free energy estimates; (2) **Structural fidelity**, ensuring that designed sequences preserve scaffold stability; (3) **Specificity**, which assesses whether designs are distinct across different antigens; and (4) **Rationality**, measuring how well designs align with natural antibody sequence distributions. We describe these measures in detail below.

- **Shape Complementarity (SC):** A geometric descriptor that quantifies how well the molecular surfaces of antibody and antigen fit together at the binding interface. High SC values indicate closer steric matching, minimized voids, and tighter residue packing, reflecting the classical "lock-and-key" principle of molecular recognition.

- **Interface Solvent-Accessible Surface Area ($\Delta$SASA):** The reduction in solvent-exposed area upon binding, computed as the change in accessible surface area between unbound and bound states. Larger $\Delta$SASA values signal more extensive burial of hydrophobic and polar residues, generally stabilizing the interaction through favorable desolvation.

- **Interface Hydrogen Bonds (Hbonds):** The number of hydrogen bonds formed across the interface. These interactions provide directionality and chemical specificity beyond steric fit, and higher counts typically indicate stronger and more selective interactions.

- **Separated Free Energy ($\Delta G_{\text{sep}}$):** An energetic estimate from Rosetta that measures the free energy difference between the bound complex and the separated antibody–antigen structures. More negative values correspond to stronger thermodynamic stability, directly complementing structural metrics with a unified energetic score.

- **Structural Fidelity (scRMSD):** A measure of backbone agreement between the original scaffold and the refolded design. Lower values indicate that designed sequences reliably fold into the intended conformation, ensuring that affinity improvements are not achieved at the expense of structural stability.

- **Specificity (Sim):** To assess whether designs are target-tailored, we compute sequence similarity across antibodies generated for different antigens. Lower similarity values reflect more antigen-distinct repertoires, discouraging convergence onto generic binding motifs.

- **Rationality / Naturalness (Nat):** Using inverse perplexity from AntiBERTy (Ruffolo et al., 2021), we quantify how natural designs appear under a pretrained antibody language model. Higher values indicate stronger alignment with evolutionary sequence constraints, which supports realistic folding, expression, and function.

Together, these metrics provide a comprehensive assessment of antibody design quality. Interface metrics (SC, $\Delta$SASA, Hbonds) and binding free energy ($\Delta G_{\text{sep}}$) are useful indicators to measure binding affinity; scRMSD ensures structural fidelity, i.e. self-consistency between initial backbone and refolded structure; Sim enforces antigen specificity; and Nat evaluates evolutionary plausibility. Note that when we report SC, $\Delta$SASA, Hbonds, scRMSD in the main table, and scRMSD and $\Delta G_{\text{sep}}$ in the Pareto front plot, we normalize them to [0, 1]. The normalizing scalers are obtained from evaluated metrics of $4,096$ seed sequences for each target antigen.

### D.5 MORE DESCRIPTIONS ON THE TARGET ANTIGENS

In this study, we focus on four representative antigen targets spanning diverse biological contexts:

- **PD1** (*Programmed Cell Death Protein 1*): PD1 is an inhibitory receptor expressed primarily on activated T cells, B cells, and natural killer cells. By binding to its ligands PD-L1 or PD-L2, PD1 downregulates immune responses to maintain self-tolerance and prevent autoimmunity. However, many tumors overexpress PD-L1 to suppress anti-tumor immunity and promote immune evasion. Therapeutically, antibodies blocking PD1 or PD-L1 have become cornerstone immunotherapies in oncology, restoring T cell activity and inducing durable tumor regression in multiple cancer types.

- **SARS-CoV-2 RBD** (*Receptor Binding Domain of the Spike Protein*): The RBD of the SARS-CoV-2 spike glycoprotein mediates viral attachment by engaging the angiotensin-converting enzyme 2 (ACE2) receptor on host cells, initiating viral entry. Because of its central role in infection, the RBD is the primary target of neutralizing antibodies elicited by natural infection and vaccination. Mutations in the RBD define many variants of concern, influencing transmissibility, immune escape, and vaccine effectiveness. As such, the RBD represents a critical antigenic target for both therapeutic antibody development and next-generation vaccine design.

- **IL7** (*Interleukin-7*): IL7 is a cytokine essential for T cell development, survival, and homeostatic proliferation. It signals through the IL7 receptor, a heterodimer composed of IL7R$\alpha$ and the common $\gamma$-chain, activating JAK/STAT pathways to regulate lymphocyte biology. Dysregulated IL7/IL7R signaling has been linked to autoimmune diseases, immunodeficiency syndromes, and

hematologic malignancies such as acute lymphoblastic leukemia. Clinically, IL7 has been explored as a therapeutic agent to enhance immune reconstitution after chemotherapy, bone marrow transplantation, or chronic viral infection, while IL7 pathway inhibition may offer strategies to treat autoimmunity.

- **INSR** (*Insulin Receptor*): The insulin receptor is a transmembrane receptor tyrosine kinase that mediates the metabolic and mitogenic effects of insulin. Upon insulin binding, INSR activates signaling cascades such as PI3K/AKT and MAPK, which regulate glucose uptake, metabolism, and cell growth. Mutations in INSR can cause severe insulin resistance syndromes, while more subtle alterations in receptor activity contribute to common metabolic disorders like type 2 diabetes. Beyond metabolism, dysregulated INSR signaling has been implicated in cancer, where it can promote proliferation and survival. Antibodies targeting INSR therefore hold potential not only for metabolic disease modulation but also as tools to interfere with oncogenic signaling pathways.

## D.6 COMPUTATIONAL COST

Table 3 summarizes the computational budget for each optimization round, including the number of evaluated sequences, wall-clock time for policy training, and per-structure evaluation time (RFdiffusion backbone generation, IgFold refolding, and Rosetta relaxation). Hardware specifications for both CPU and GPU resources used in our experiments are also provided.

Table 3: Summary of computation time and hardware specifications per optimization round.

| Metric | Value |
|---|---|
| # of evaluated sequences | 4096 |
| Training time per epoch | 231.4 s |
| Refolding time per structure (IgFold) | 30.5 s |
| Relaxation time per structure (Rosetta) | 9.4 s |
| Backbone generation time (RFdiffusion) | 14.5 s |
| Hardware specification (CPU) | AMD EPYC 7R13 |
| Hardware specification (GPU) | NVIDIA L40S |

## E ALGORITHMIC DESCRIPTION

We summarize the AbMPO procedure given in Algorithm 1, highlighting its core components and rationale.

**Overview**   AbMPO is an iterative policy optimization framework tailored for antibody sequence design. The algorithm integrates (i) offline reward modeling using surrogate structural metrics and (ii) on-policy refinement with KL-regularized regression. This two-stage structure balances the limited availability of high-quality affinity labels with the need for stable and sample-efficient policy improvement.

**Inputs and Initialization**   The algorithm requires:

- A training context set $D$, where each context $C$ represents an antigen/backbone prompt.
- A pretrained reference policy $\pi_{\mathrm{ref}}$, serving as the initialization $\pi_0$.
- Hyperparameters: temperature $\beta$ (controls softness of value estimates), offline batch size $N_{\mathrm{ref}}$, per-iteration on-policy batch size $B$, and total number of optimization rounds $T$.

The reference policy anchors training, while the temperature parameter tunes the trade-off between exploration and stability.

**Offline Soft-Value Estimation**   For each context $C$, a batch of $N_{\mathrm{ref}}$ candidate sequences $\{S_i\}$ is sampled from the reference policy. Each sequence is structurally evaluated (e.g., refolding followed

by Rosetta scoring) to compute a surrogate reward $r(C, S_i)$. Using these scores, a *soft value estimate* is computed as

$$\hat{V}(C) \;=\; \beta \log \left( \tfrac{1}{N_{\text{ref}}} \sum_{i=1}^{N_{\text{ref}}} \exp\!\big(r(C, S_i)/\beta\big) \right).$$

This estimator corresponds to a log-sum-exp average of rewards and provides a smooth baseline for variance reduction. Importantly, this stage is entirely offline with respect to $\pi_{\text{ref}}$, ensuring stability before iterative updates.

**On-Policy Refinement**   At each iteration $t$, the current policy $\pi_t$ is refined using freshly sampled sequences from $\pi_t(\cdot \mid C)$. For each candidate sequence $S$, the advantage is estimated as

$$\hat{A}_t(C, S) = r(C, S) - \hat{V}_t(C),$$

where $\hat{V}_t(C)$ may be recomputed using the on-policy batch.

The policy is then updated by minimizing a *KL-stabilized regression loss*:

$$\mathcal{L}_t(\pi_\theta) \;=\; \mathbb{E}_{C \sim D}\, \mathbb{E}_{S \sim \pi_t(\cdot|C)} \Big[ \big(\beta \log \tfrac{\pi_\theta(S|C)}{\pi_t(S|C)} - \hat{A}_t(C, S)\big)^2 \Big].$$

This regression formulation can be interpreted as aligning the policy gradient with a KL-regularized reward maximization objective, while stabilizing updates through squared-error regression. The KL term acts as a trust region, preventing destabilizing jumps in policy space.

**Iterative Loop**   The refinement loop is repeated for $T$ iterations, producing policies $\pi_1, \pi_2, \ldots, \pi_T$. Early stopping or additional trust-region constraints can be applied if the KL divergence between successive policies exceeds a threshold. The final learned policy $\pi_T$ is returned.

**Notes**

1. **Reward Design.** The reward function $r(C, S)$ is a surrogate affinity score, typically defined as a weighted combination of Rosetta-derived interface metrics (shape complementarity, buried surface area, hydrogen bonds, etc.). The algorithm is agnostic to the specific form of the reward, making it extensible to other task-specific metrics.

2. **Offline/Online Balance.** Offline soft-value estimation ensures stable baselines, while on-policy refinement allows continuous adaptation of the policy toward higher-affinity solutions.

3. **KL Regularization.** By grounding each update in a KL divergence to the previous policy, the method avoids catastrophic policy collapse and ensures smooth optimization dynamics.

# F   ADDITIONAL EXPERIMENTAL RESULTS

## F.1   MORE ANALYSIS ON BENCHMARK COMPARISON

In Table 4, we present the benchmark results on two target antigens: PD1 and SARS-CoV-2 RBD. We make the following observations: (1) Our method (AbMPO) achieves the best overall performance across PD1 and SARS-CoV-2 RBD, consistently ranking first in SC, $\Delta$SASA, hbonds, and $\Delta G_{\text{sep}}$, confirming that the surrogate reward effectively drives improvements in binding affinity. (2) AbMPO also delivers the lowest Sim scores, demonstrating its ability to generate antigen-specific and diverse sequences, a key advantage of structure-conditioned optimization compared to language-only methods, which lack explicit antigen conditioning. (3) In terms of Nat, AbMPO achieves the highest score overall, surpassing even language model baselines despite their pretraining on large antibody corpora. This indicates that structure-conditioned sequence design combined with policy optimization can yield both natural and high-affinity sequences. (4) Among baselines, Ab-DPO achieves strong second-best results on several affinity-related metrics, reflecting the benefit of energy-aware preference optimization. Similarly, AbMPNN performs competitively on SC and scRMSD, showing the benefit of antibody-specific adaptation within structure-conditioned models.

In Table 5, we report results on two additional antigen targets, IL7 and INSR, across the same seven evaluation metrics. We make the following observations: (1) AbMPO achieves the best overall

---

**Algorithm 1** Iterative Policy Optimization for Antibody Sequence Design (AbMPO)

---

**Require:** Training context set $D$ (antigen/backbone prompts $C$), reference policy $\pi_{\text{ref}}$, initial policy $\pi_0 \leftarrow \pi_{\text{ref}}$, temperature $\beta$, offline batch size $N_{\text{ref}}$, per-iteration batch size $B$, iterations $T$

**Ensure:** Learned policy $\pi_T$

    `# Iterative On-Policy Refinement with KL-Stabilized Regression`

1: **for** $t = 0$ **to** $T - 1$ **do**

    `# Offline Soft-Value Estimation under` $\pi_{\text{ref}}$

2:     **for all** $C \in D$ **do**

3:         Draw $N_{\text{ref}}$ i.i.d. sequences $\{S_i\}_{i=1}^{N_{\text{ref}}} \sim \pi_{\text{ref}}(\cdot \mid C)$

4:         For each $S_i$, run structure prediction and Rosetta to obtain reward $r(C, S_i)$

5:         Compute soft value estimate

$$\hat{V}(C) \;\leftarrow\; \beta \log\left(\frac{1}{N_{\text{ref}}} \sum_{i=1}^{N_{\text{ref}}} \exp\big(r(C, S_i)/\beta\big)\right)$$

6:     **end for**

7:     $\hat{A}_t(C, S_j) \leftarrow r(C, S_j) - \hat{V}_t(C)$

8:     **Policy Update:** update $\pi_{t+1}$ by (stochastic) gradient descent on

$$\mathcal{L}_t(\pi_\theta) \;=\; \mathbb{E}_{C \sim D} \, \mathbb{E}_{S \sim \pi_t(\cdot|C)}\left[\big(\beta \log \tfrac{\pi_\theta(S|C)}{\pi_t(S|C)} \;-\; \hat{A}_t(C, S)\big)^2\right]$$

9: **end for**

10: **Return** $\pi_T$

---

    **Notes.**

    (i) Rewards $r(C, S)$ are the surrogate affinity scores (e.g., a weighted combination of SC, $\Delta$SASA, Hbonds), but the algorithm is agnostic to the exact choice.

    (ii) Soft-value estimation is fully offline under $\pi_{\text{ref}}$; The overall algorithm loop performs on-policy refinement with KL regularization via the regression loss.

---

Table 4: Overall performance on PD1 and SARS-CoV-2 RBD across different baselines. Best results are shown in **bold** and second-best results are underlined.

| Method | PD1 | | | | | | | SARS-CoV-2 RBD | | | | | | |
|---|---|---|---|---|---|---|---|---|---|---|---|---|---|---|
| | SC↑ | ΔSASA↑ | Hbonds↑ | $\Delta G_{\text{sep}}$↓ | scRMSD↓ | Sim↓ | Nat↑ | SC↑ | ΔSASA↑ | Hbonds↑ | $\Delta G_{\text{sep}}$↓ | scRMSD↓ | Sim↓ | Nat↑ |
| ESM-IF | 0.374 | 0.147 | 0.007 | 242.213 | 0.624 | 0.576 | 0.351 | 0.382 | 0.141 | 0.007 | 184.671 | 0.606 | 0.571 | 0.344 |
| AbLang | 0.405 | 0.153 | 0.009 | 228.371 | 0.581 | 0.534 | 0.337 | 0.413 | 0.147 | 0.008 | 174.425 | 0.564 | 0.529 | 0.354 |
| nanoBERT | 0.378 | 0.142 | 0.008 | 236.912 | 0.603 | 0.538 | 0.328 | 0.378 | 0.135 | 0.007 | 180.632 | 0.583 | 0.536 | 0.343 |
| InstructPLM | 0.427 | 0.151 | 0.008 | 205.219 | 0.572 | 0.543 | 0.364 | 0.436 | 0.146 | 0.009 | 152.314 | 0.539 | 0.545 | 0.358 |
| DiffAb | 0.408 | 0.159 | 0.008 | 225.191 | 0.567 | 0.545 | 0.312 | 0.423 | 0.158 | 0.008 | 168.149 | 0.527 | 0.551 | 0.319 |
| AbDPO | 0.412 | 0.164 | 0.010 | 178.582 | 0.557 | 0.541 | 0.319 | 0.427 | 0.163 | 0.010 | 132.857 | 0.518 | 0.542 | 0.331 |
| ProteinMPNN$_{\text{AR}}$ | 0.382 | 0.143 | 0.007 | 212.344 | 0.628 | 0.554 | 0.344 | 0.387 | 0.141 | 0.007 | 160.734 | 0.591 | 0.551 | 0.344 |
| ProteinMPNN$_{\text{CMLM}}$ | 0.395 | 0.149 | 0.008 | 198.763 | 0.604 | 0.546 | 0.338 | 0.399 | 0.147 | 0.008 | 152.582 | 0.589 | 0.539 | 0.346 |
| AbMPNN | 0.434 | 0.173 | 0.009 | 133.562 | 0.579 | **0.525** | 0.349 | 0.432 | 0.174 | 0.009 | 121.364 | 0.527 | 0.527 | 0.360 |
| **AbMPO** | **0.451** | **0.203** | **0.013** | **119.326** | 0.556 | 0.531 | **0.367** | **0.456** | **0.192** | **0.012** | **90.932** | **0.509** | 0.525 | **0.378** |

performance, consistently ranking first in SC, $\Delta$SASA, Hbonds, and $\Delta G_{\text{sep}}$ on both antigens, confirming that the surrogate reward drives improvements in binding affinity. (2) AbMPO also yields the lowest Sim scores, highlighting its ability to produce antigen-specific and diverse sequences, while achieving the highest Nat scores, surpassing even pretrained language model baselines. (3) Among the baselines, InstructPLM and AbMPNN are the strongest competitors, with InstructPLM achieving competitive SC and $\Delta G_{\text{sep}}$ values, and AbMPNN performing well on $\Delta$SASA, Hbonds, and Nat. (4) AbDPO stands out for its superior scRMSD, reflecting its strength in preserving structural fidelity, though it lags behind AbMPO in affinity-related metrics.

### F.2 VALIDATION OF SURROGATE REWARD DESIGN

To validate our surrogate reward formulation, we performed a linear regression analysis of experimental $\Delta\Delta G$ values against four Rosetta-derived interface metrics: *shape complementarity* (SC), *interface solvent-accessible surface area* ($\Delta$SASA), *interface hydrogen bonds* (Hbonds), and the

Table 5: Overall performance on IL7 and INR across different baselines. Best results are shown in **bold** and second-best results are underlined.

| Method | IL7 | | | | | | | INR | | | | | | |
|---|---|---|---|---|---|---|---|---|---|---|---|---|---|---|
| | SC↑ | ΔSASA↑ | Hbonds↑ | $\Delta G_{\text{sep}}$↓ | scRMSD↓ | Sim↓ | Nat↑ | SC↑ | ΔSASA↑ | Hbonds↑ | $\Delta G_{\text{sep}}$↓ | scRMSD↓ | Sim↓ | Nat↑ |
| ESM-IF | 0.365 | 0.136 | 0.007 | 262.114 | 0.628 | 0.572 | 0.331 | 0.371 | 0.133 | 0.007 | 219.776 | 0.611 | 0.568 | 0.328 |
| AbLang | 0.394 | 0.147 | 0.009 | 247.331 | 0.592 | 0.530 | 0.342 | 0.401 | 0.143 | 0.009 | 198.155 | 0.576 | 0.536 | 0.347 |
| nanoBERT | 0.371 | 0.138 | 0.008 | 257.022 | 0.615 | 0.534 | 0.323 | 0.370 | 0.135 | 0.008 | 205.276 | 0.593 | 0.536 | 0.339 |
| InstructPLM | 0.419 | 0.146 | 0.008 | 224.532 | 0.582 | 0.538 | 0.357 | 0.423 | 0.144 | 0.009 | 174.832 | 0.549 | 0.542 | 0.354 |
| DiffAb | 0.400 | 0.153 | 0.008 | 245.129 | 0.577 | 0.540 | 0.307 | 0.415 | 0.152 | 0.008 | 192.251 | 0.537 | 0.547 | 0.315 |
| AbDPO | 0.404 | 0.158 | **0.010** | 204.789 | 0.568 | 0.536 | 0.314 | 0.419 | 0.159 | 0.010 | 182.893 | **0.526** | 0.538 | 0.327 |
| ProteinMPNN$_{\text{AR}}$ | 0.374 | 0.138 | 0.007 | 232.849 | 0.636 | 0.550 | 0.338 | 0.379 | 0.137 | 0.007 | 187.324 | 0.601 | 0.547 | 0.340 |
| ProteinMPNN$_{\text{CMLM}}$ | 0.387 | 0.143 | 0.008 | 217.908 | 0.612 | 0.543 | 0.332 | 0.392 | 0.142 | 0.008 | 178.215 | 0.597 | 0.535 | 0.343 |
| AbMPNN | 0.416 | 0.156 | 0.008 | 221.209 | 0.583 | 0.522 | 0.366 | 0.408 | 0.161 | 0.013 | 185.969 | 0.598 | 0.542 | 0.361 |
| **AbMPO** | **0.439** | **0.167** | **0.010** | **202.184** | **0.566** | **0.518** | **0.384** | **0.428** | **0.179** | **0.015** | **174.559** | 0.574 | **0.531** | **0.373** |

Rosetta binding energy estimate ($\Delta G_{\text{sep}}$). The regression coefficients, standard errors, test statistics, and variance inflation factors (VIF) are summarized in Table 6.

Table 6: Regression of experimental $\Delta\Delta G$ values against four Rosetta-derived interface metrics.

| Metric | Coefficient | Std. Error | t-Statistic | $p$-value | VIF |
|---|---|---|---|---|---|
| $\Delta G_{\text{sep}}$ | $-0.90$ | 2.01 | $-0.44$ | 0.657 | 1.03 |
| ΔSASA | $-38.46$ | 2.91 | $-13.23$ | $2.1 \times 10^{-39}$ | 1.14 |
| Hbonds | $-10.97$ | 1.98 | $-5.54$ | $3.1 \times 10^{-8}$ | 1.12 |
| SC | $-12.95$ | 0.97 | $-13.30$ | $8.3 \times 10^{-40}$ | 1.01 |

The coefficient of $\Delta G_{\text{sep}}$ is close to zero and not statistically significant ($p = 0.66$), indicating that it does not explain meaningful variance in experimental $\Delta\Delta G$. Variance inflation factors are close to 1 for all predictors, ruling out multicollinearity. In contrast, SC, ΔSASA, and Hbonds exhibit large and statistically significant negative coefficients, consistent with the expectation that higher structural complementarity, buried surface area, and interfacial hydrogen bonding correspond to stronger binding affinity (i.e., lower $\Delta\Delta G$).

Based on these results, we refit the regression using only SC, ΔSASA, and Hbonds. The updated coefficients are shown in Table 7. Excluding $\Delta G_{\text{sep}}$ yields a more stable and interpretable model, with coefficients of similar magnitude and direction.

Table 7: Refitted regression excluding $\Delta G_{\text{sep}}$.

| Metric | Coefficient | Std. Error | t-Statistic | $p$-value | VIF |
|---|---|---|---|---|---|
| ΔSASA | $-38.26$ | 2.87 | $-13.32$ | $6.9 \times 10^{-40}$ | 1.11 |
| Hbonds | $-11.05$ | 1.97 | $-5.61$ | $2.1 \times 10^{-8}$ | 1.12 |
| SC | $-12.97$ | 0.97 | $-13.34$ | $5.2 \times 10^{-40}$ | 1.01 |

All coefficients remain negative, confirming that increases in SC, ΔSASA, and Hbonds are associated with reductions in experimental $\Delta\Delta G$. For clarity in constructing the surrogate reward, we therefore take the absolute values of these coefficients as weights, ensuring that the weighted surrogate reward is aligned with improvements in binding affinity. This weighted combination effectively captures the same predictive signal as $\Delta G$, which in turn approximates $\Delta\Delta G$.

### F.3    STATISTICS OF METRICS AND REWARDS

Here we provide the histograms and distributions of three different metrics and the surrogate reward at the first round of policy optimization training for PD1 antigen. The results are shown in Figure 5.

### F.4    DISTRIBUTION SHIFT IN TRAINING

Here, we provide additional visualizations to highlight the distributional shifts observed in three key interface metrics—shape complementarity (SC), solvent-accessible surface area reduction

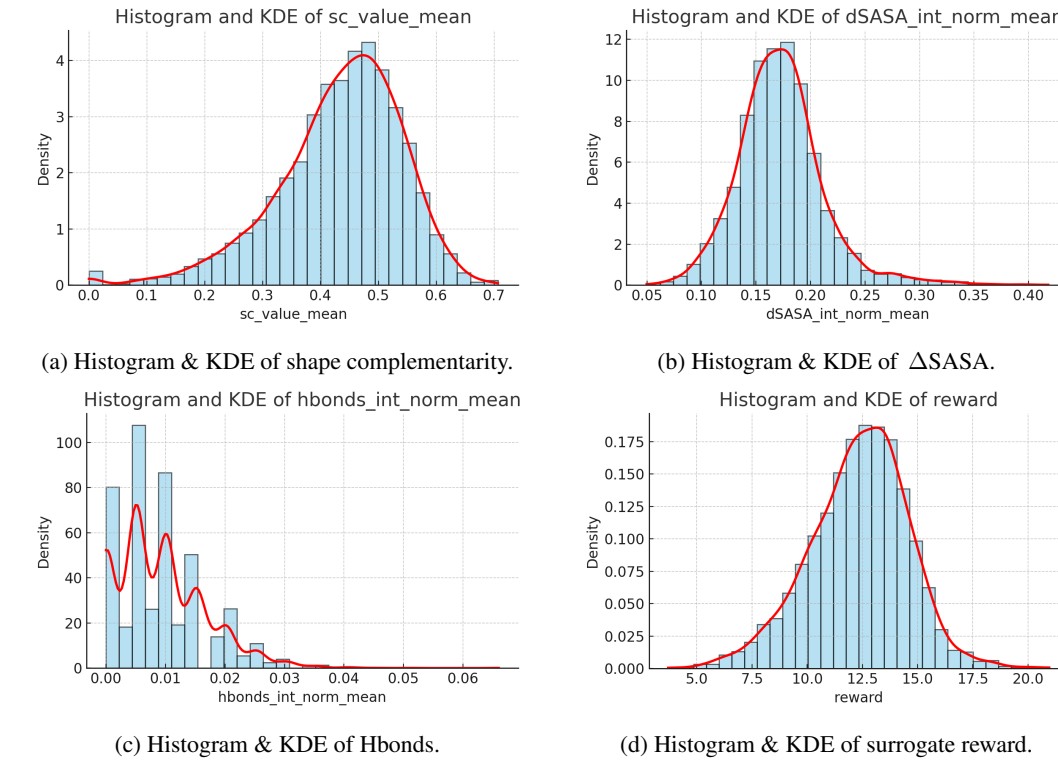

(a) Histogram & KDE of shape complementarity.

(b) Histogram & KDE of $\Delta$SASA.

(c) Histogram & KDE of Hbonds.

(d) Histogram & KDE of surrogate reward.

Figure 5: Distributions (histogram + KDE) for the three metrics and the surrogate reward.

($\Delta$SASA), and the number of hydrogen bonds—as well as in the surrogate reward function. These results are shown after one round of policy optimization training and capture the differences between baseline and optimized models. To ensure a comprehensive evaluation, we present these distributions for three representative target antigens—SARS-CoV-2 RBD, IL7, and INR. The case of PD1 is omitted here, as its corresponding results have already been presented and discussed in the main text. Together, these visualizations further demonstrate how policy optimization shifts the underlying design distribution toward improved structural and energetic properties across diverse antigens. The results are shown in Figure 6, Figure 7, and Figure 8.

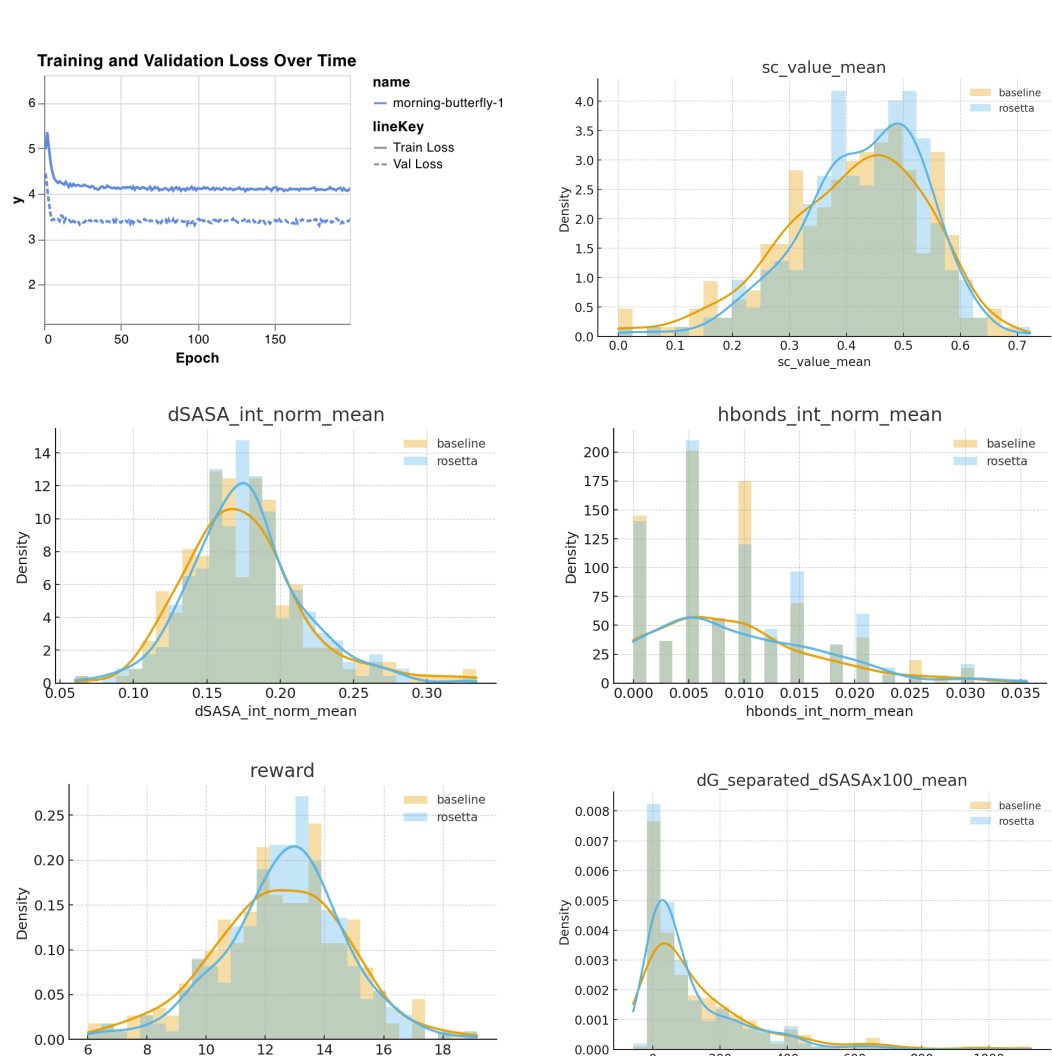

Figure 6: Training and evaluation results. (Top left) Training and validation loss curves over epochs. (Top right) Distribution of shape complementarity (SC) values. (Middle row) Normalized $\Delta$SASA and interface H-bonds distributions. (Bottom row) Reward distributions and $\Delta G_{\text{sep}}$ values. Comparison is shown between AbMPNN (*baseline*) and AbMPO (*rosetta*) on SAR-CoV-2 RBD.

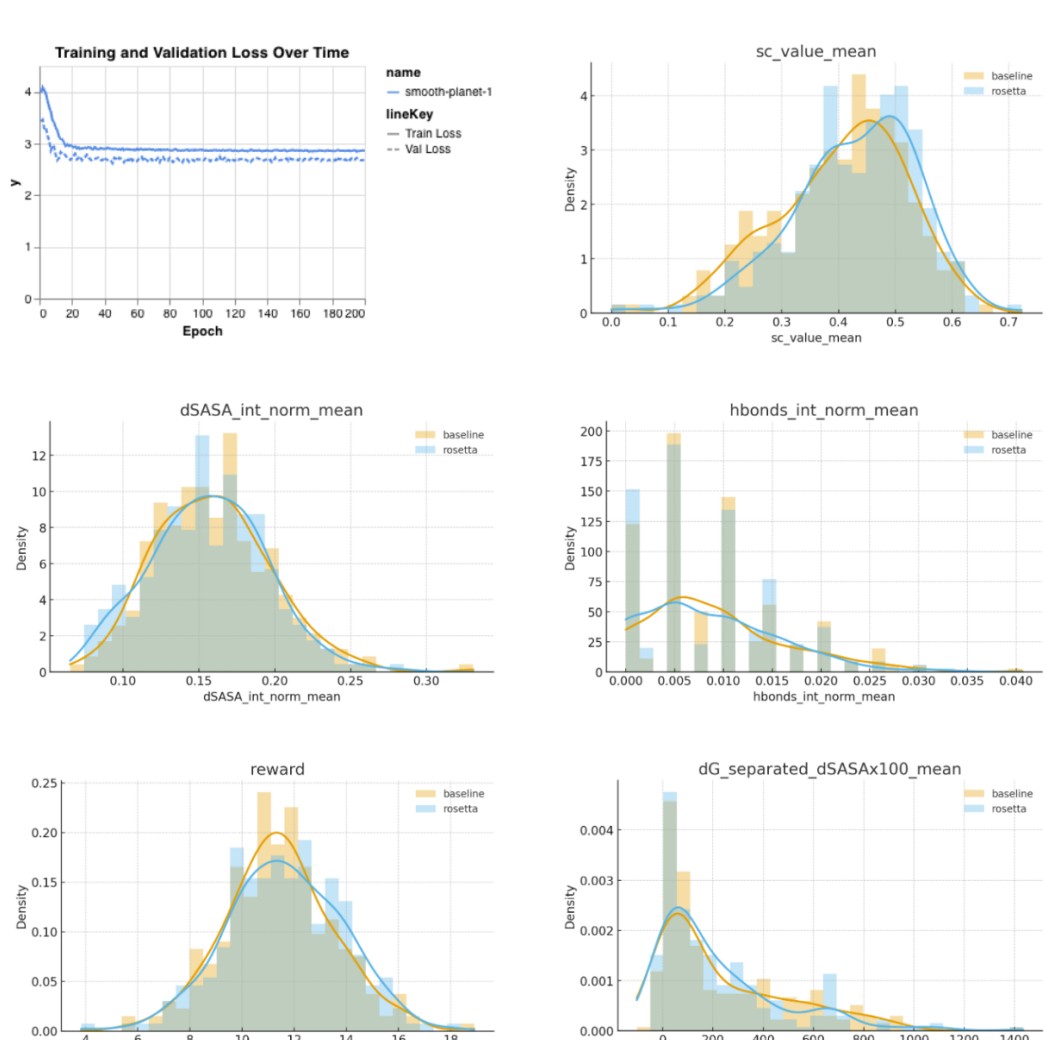

Figure 7: Training and evaluation results. (Top left) Training and validation loss curves over epochs. (Top right) Distribution of shape complementarity (SC) values. (Middle row) Normalized $\Delta$SASA and interface H-bonds distributions. (Bottom row) Reward distributions and $\Delta G_{\text{sep}}$ values. Comparison is shown between AbMPNN (*baseline*) and AbMPO (*rosetta*) on IL7.

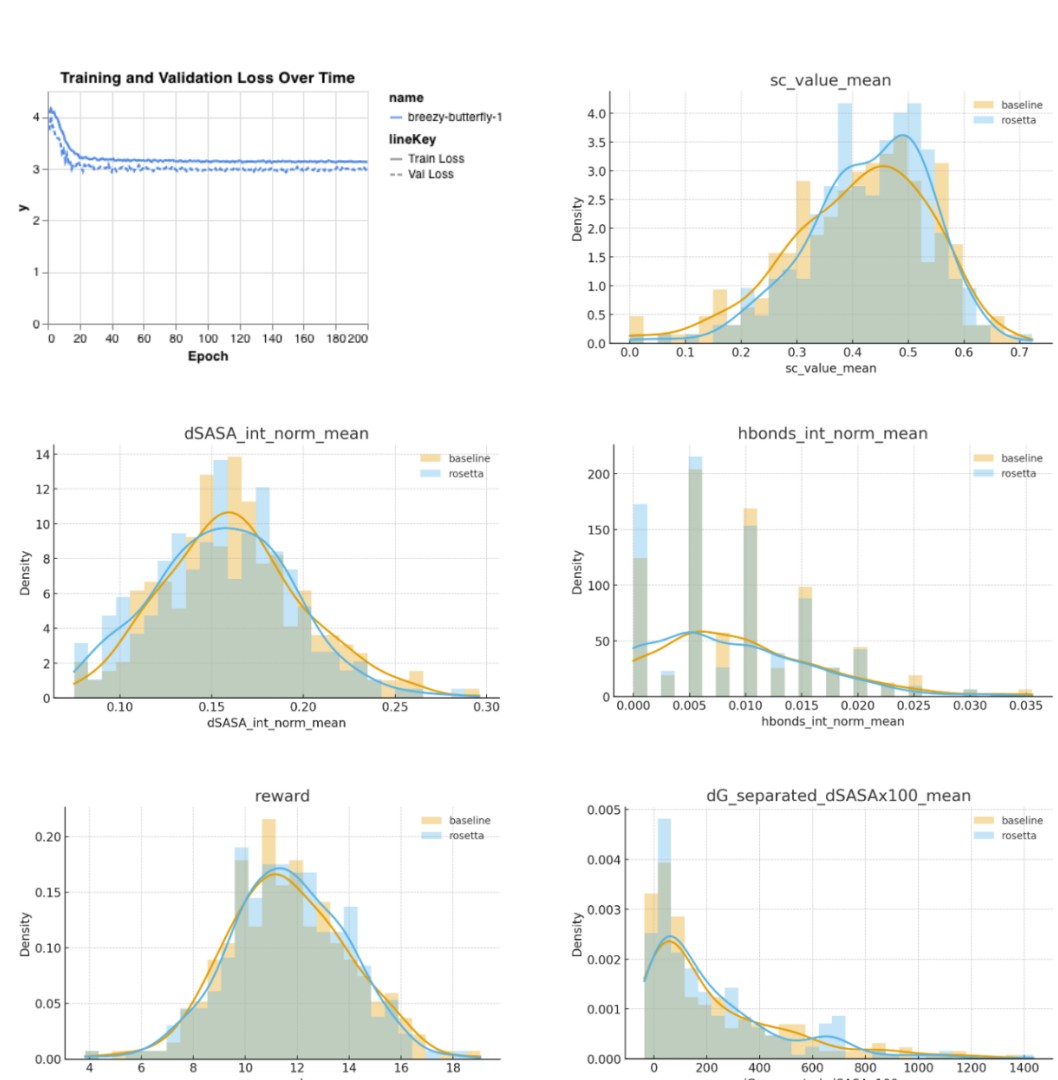

Figure 8: Training and evaluation results. (Top left) Training and validation loss curves over epochs. (Top right) Distribution of shape complementarity (SC) values. (Middle row) Normalized $\Delta$SASA and interface H-bonds distributions. (Bottom row) Reward distributions and $\Delta G_{\text{sep}}$ values. Comparison is shown between AbMPNN (*baseline*) and AbMPO (*rosetta*) on INR.

