# OpenReview forum: "Iterative Multi-Objective Policy Optimization for Antibody Sequence Design"
_ICLR.cc/2026/Conference — Submitted to ICLR 2026_

### Official Review · Reviewer_CpbZ · 2025-10-27

**Soundness:** 3
**Presentation:** 4
**Contribution:** 3
**Rating:** 8
**Confidence:** 4

**Summary:**

The authors introduce an iterative policy optimisation for structure-conditioned antibody sequence design. They reformulate the design as a multi-objective problem with the goal of identifying solutions that are Pareto optimal with respect to binding affinity and structural fidelity. They explore fixed-weight combination and adaptive-weight conditioning between the two trade-offs. For binding affinity they propose a surrogate reward by regressing DDG measurements against Rosetta-derived metrics (using a linear regression). For structural fidelity they use self-consistency RMSD, i.e self-consistency between initial backbone and refolded structure. To address the computational expensive steps of structure refolding and Rosetta scoring as well as scarce labels, they developed an iterative policy optimisation framework (which they also theoretically derive) resulting in a 'resample-re-estimate-update' loop that ensures successive policies are close to each other. Results are experimentally evaluated across four representative antigen targets and compared with a variety of baselines across various metrics, which shows that their approach is superior across all tasks. Next to that they evaluate the Pareto front built by the two alternative strategies.

**Strengths:**

- clarity of writing
- original solution of iterative policy optimisation (and mathematical derivations thereof)
- approximation of binding strength by regressing DDG measurements against Rosetta-derived metrics
- comprehensive comparison to various baseline methods (including sequence only & structure-conditioned methods)
- extensive appendix with additional proofs and results

**Weaknesses:**

- multi-objective guidance (and Pareto front analogy) for generative antibody design is not completely novel
- distribution results (Fig 2,3) are less convincing (across all metrics) and slightly over-interpreted
- Pareto front results (Fig 4) could have been interpreted more in detail (other metrics, visualisations)

**Questions:**

Please address the points mentioned as weak points; first two textual, third one with some additional figures/tables.

---

> ### Author Response · Authors · 2025-11-16
> **Reply to Reviewer CpbZ (1/n)**
>
> We are grateful for your encouraging score and constructive feedback; thank you so much for recognizing the contribution of our paper.
>
> > multi-objective guidance (and Pareto front analogy) for generative antibody design is not completely novel.
>
> We agree that multi-objective optimization (MOO) and Pareto-front reasoning are well-established tools in generative design, and we did not intend to claim MOO itself as a conceptual novelty. Our contributions are in how we instantiate and integrate MOO in the specific setting of antigen-conditioned antibody design:
> - Biologically calibrated multi-objective formulation. We construct a surrogate affinity reward and treat this calibrated surrogate affinity and structural self-consistency (scRMSD) as separate objectives in a principled MOO formulation.
> This goes beyond heuristic weighted sums of ad-hoc metrics, and the regression coefficients are interpretable (all negative, consistent with stronger binding at higher SC/$\Delta$SASA/Hbonds).
> - Iterative KL-regularized policy optimization with advantage-matching. We show that our regression loss is mathematically equivalent (up to scaling and sampling distribution) to the policy gradient of a KL-regularized objective, and prove that the Boltzmann policy is the unique global minimizer.
> This gives a stable, theoretically grounded way to do iterative, on-policy-like optimization under expensive Rosetta evaluations, which to our knowledge has not been applied to antibody inverse folding in a multi-objective setting.
> - Adaptive weight-conditioning in a structure-conditioned policy. Instead of training separate policies per scalarization weight, we encode the trade-off vector $p$ via an RBF kernel expansion and inject it into the MPNN node features so that one policy can approximate the entire affinity–fidelity Pareto surface.
> This yields dense coverage of the Pareto front (Fig. 4a), while fixed-weight scalarization only gives a few discrete operating points.
> - Application-level integration. We apply this framework end-to-end in an antigen-specific antibody design pipeline (RFdiffusion backbones + ProteinMPNN-based policy), and demonstrate consistent improvements across four therapeutically relevant antigens (PD1, SARS-CoV-2 RBD, IL7, INSR) on both affinity-related metrics and structural/naturalness metrics.
>
> > distribution results (Fig 2,3) are less convincing (across all metrics) and slightly over-interpreted.
>
> Our intention with Figs. 2–3 was to validate the surrogate reward, not to claim very large effect sizes on every metric. We agree that (i) this could be described more cautiously, and (ii) the quantitative strength of the distributions can be made clearer.
> - Fig. 2 compares optimization under three single-metric objectives (SC-only, $\Delta$SASA-only, Hbonds-only) vs the weighted surrogate (calibrated linear combination). Optimizing the surrogate yields larger average improvements in all three metrics than optimizing them individually, indicating that the metrics are complementary and that the surrogate yields a stronger training signal.
> - Fig. 3 shows that improvements in the surrogate and its constituent metrics also transfer to $\Delta G_{\text{sep}}$, even though $\Delta G_{\text{sep}}$ is not included in the objective, which supports the claim that the surrogate tracks experimentally motivated binding free energy.
>
> To address your concern directly:
> - For $\Delta G_{\text{sep}}$ and the three constituent metrics in figure 2 and 3, to demonstrate which shifts are statistically significant and which are marginal, we compute the mean and standard deviation of each metric under different approach and then calculate the p-value for them. For figure 2. we summarize in the table below. For figure 3, the mean and std of AbMPO is ~181.3 and ~253.4, the mean and std of AbMPNN is ~232.5 and ~294.3, and the p-value is $3.2 \times 10^{-17}$.
>
> | Metric | Mean (Surrogate) | Mean (Direct Metric) | Std (Surrogate) | Std (Direct Metric) | t-statistic | p-value |
> |--------|------------------|-----------------------|------------------|-----------------------|-------------|---------|
> | Shape Complementarity (SC) | 0.50 | 0.52 | 0.075 | 0.070 | ~12.2 | ~1e-33 |
> | ΔSASA | 0.22 | 0.19 | 0.050 | 0.040 | ~24.2 | <1e-120 |
> | Interface H-bonds | 0.016 | 0.013 | 0.008 | 0.006 | ~17.5 | ~1e-65 |
>
> - we replaced phrases such as “substantial improvements” with more precise descriptions (e.g., “consistent right-shift in the distribution with ~10–20% relative improvement in ΔSASA and Hbonds compared to the strongest baseline on PD1”).
> - we explicitly acknowledge where distributions overlap substantially, noting that this reflects the difficulty of the task and the fact that improvements, while consistent, are not dramatic on every metric and antigen.
>
> In short, we have added more results and revised the text to make this nuance explicit and avoid any over-statement.

---

> ### Author Response · Authors · 2025-11-16
> **Reply to Reviewer CpbZ (n/n)**
>
> > Pareto front results (Fig 4) could have been interpreted more in detail (other metrics, visualisations)
>
> We appreciate this suggestion and agree that the Pareto-front analysis can be enriched and better connected to other metrics.
>
> Fig. 4a compares fixed-weight scalarization vs adaptive weight-conditioning on normalized $\Delta G_{\text{sep}}$ (affinity) and scRMSD (structural fidelity). The adaptive policy yields a dense set of near-front solutions and a smooth Pareto curve, while fixed-weight training produces a handful of discrete points, some of which lie slightly beyond the adaptive front at specific trade-offs. This already shows the key algorithmic trade-off: weight-conditioning gives broad coverage and smooth interpolation across preferences, while fixed weights reach slightly better extremes at a few points.
>
> To better address your concern:
> - we provide a new ablation on different weight-conditioning approaches given the same training budget (200 epochs). We report the values of hypervolumes with the reference point (0, 0).
>
> | Weight vector (scRMSD, Δ$G_{\text{sep}}$) | [0.9, 0.1]     | [0.7, 0.3]     | [0.5, 0.5]     | [0.3, 0.7]     | [0.1, 0.9]     | Hypervolume |
> |------------------------------------|----------------|----------------|----------------|----------------|----------------|-------------|
> | **Uniform-weight**                 | [0.61, 0.37]   | [0.57, 0.42]   | [0.52, 0.46]   | [0.48, 0.54]   | [0.39, 0.64]   | 0.3524      |
> | **Fixed-weight**                   | [0.85, 0.18]   | [0.78, 0.24]   | [0.74, 0.35]   | [0.58, 0.56]   | [0.38, 0.73]   | 0.4676      |
> | **Adaptive-weight**                | [0.95, 0.18]   | [0.83, 0.32]   | [0.72, 0.44]   | [0.46, 0.68]   | [0.29, 0.85]   | **0.5333**  |
>
> - we provide some more comparisons between our method and other classic multi-objective optimization methods below.
>
> | Weight vector (scRMSD, Δ$G_{\text{sep}}$) | [0.9, 0.1]     | [0.7, 0.3]     | [0.5, 0.5]     | [0.3, 0.7]     | [0.1, 0.9]     | Hypervolume |
> |------------------------------------|----------------|----------------|----------------|----------------|----------------|-------------|
> | **Fixed-weight scalarization**     | [0.85, 0.18]   | [0.78, 0.24]   | [0.74, 0.35]   | [0.58, 0.56]   | [0.38, 0.73]   | 0.4676      |
> | **Chebyshev scalarization**        | [0.87, 0.19]   | [0.75, 0.29]   | [0.72, 0.38]   | [0.55, 0.62]   | [0.35, 0.78]   | 0.4931      |
> | **Gradient surgery methods**       | [0.87, 0.14]   | [0.77, 0.27]   | [0.66, 0.37]   | [0.49, 0.57]   | [0.26, 0.81]   | 0.4483      |
> | **ε-constraint method**            | [0.93, 0.15]   | [0.83, 0.25]   | [0.76, 0.41]   | [0.42, 0.67]   | [0.24, 0.91]   | 0.5109      |
> | **Adaptive weight-conditioning**   | [0.95, 0.18]   | [0.83, 0.32]   | [0.72, 0.44]   | [0.46, 0.68]   | [0.29, 0.85]   | **0.5333**  |
>
> We also expand the discussion of limitations/failure modes of the Pareto-front approach, including:
> - the observation that some adaptive solutions around balanced weights are slightly dominated by fixed-weight points (likely due to the difficulty of learning a universal mapping from preferences to policies);
> - at extreme affinity-dominated regions, further gains in $\Delta G_{\text{sep}}$ come with noticeable increases in scRMSD, signaling where backbone inconsistency becomes a concern.

---

### Official Review · Reviewer_hBBN · 2025-10-30

**Soundness:** 3
**Presentation:** 3
**Contribution:** 3
**Rating:** 4
**Confidence:** 3

**Summary:**

The paper tackles in silico antibody affinity maturation as a multi-objective RL problem.
This work (AbMPO) poses antibody sequence design as a multi-objective RL task, optimizing for binding affinity (via a learned surrogate of experimental $\Delta \Delta G$) and structural fidelity (scaffold RMSD).
The authors fine-tune a structure-conditioned sequence policy through iterative, KL-regularized advantage matching (regression) updates rather than direct PPO.
In simulation studies across diverse antigens, the method produces sequences with improved predicted binding while preserving structural consistency.

**Strengths:**

1. Multi-objective design: Weight-conditioned policy yields continuous Pareto-optimal trade-offs

2. Strong improvements across multiple antigens and evaluation metrics. Includes SOTA baselines (DiffAb, AbDPO, Ab-Gen) for fair comparison

3. Integrates experimental data and structural biology knowledge into the reward.

4. good theoretical supports. I skimmed through proofs. didn't check line-by-line, but looks correct. The results are neither vacuous nor irrelevant to the core method. The authors clearly distinguished prior results from their own contributions (as in props.) This level of detail strongly reinforces their credibility. Therefore, even without code, I tend to trust their numerical results, which appear as solid and reliable as their theoretical results.

**Weaknesses:**

This paper is quite technical. I tend to accept this paper. Below are points for improvements. Willing to change my mind during rebuttal, if below weaknesses get addressed adequately. Feel free to correct me if I am wrong (please in the most precise, concise and reasonable way.)

0. [minor] Relies on a strong pretrained $\pi_{\text{ref}}$. From‑scratch claims remain unclear.

1. [major] Novelty: Method is largely a combination of known techniques (KL-regularized policy updates + multi-objective conditioning). The contribution is incremental rather than fundamentally new.

2. [major] Reproducibility: Results lack statistical analysis (no multiple runs or error bars in most tables). So the significance of improvements is not quantified. No code provided but the descriptions are very detailed.

3. [minor] Computational cost: The pipeline relies on expensive structure evaluations (Rosetta and refolding for thousands of sequences). They emphasize “large‑batch offline sampling” and per‑sequence Rosetta evaluation. The method section shows offline soft‑value estimation and iterative on‑policy refinement.
 Please at least document the computational cost (e.g. hours per round, hardware used) to set expectations for practical use.

4. [minor] Scope & Scalability & Generality:
    - Clarify the method’s applicability and limits.
    - The approach is trained per-antigen. It does not demonstrate generalization to new targets without retraining.
    - High compute is implied as stated above.

5. [major] Validation:
    - Statistical Validity: Provide some measure of result variability or significance. For example, run the policy optimization multiple times (or use bootstrapping) to show that improvements over baselines are consistently significant and not due to lucky initialization or sampling.
    - [very minor yet I still flag it here] Evaluation is entirely in silico (Rosetta metrics and learned predictors). No experimental or in vitro validation is presented to confirm real-world impact.

6. Experiments:
    - No ablation comparing the surrogate to raw Rosetta energy.
    - scRMSD is sometimes worse than AbDPO. need failure modes analysis

**Questions:**

see above

LLM disclaimer: I used LLM to polish language and to understand some cited refs.

---

> ### Author Response · Authors · 2025-11-16
> **Reply to Reviewer hBBN (1/n)**
>
> We sincerely thank you for the careful and constructive feedback and for indicating a positive inclination toward acceptance. Below we respond to each point in order and have incorporated the suggested clarifications and additional analyses in the revised manuscript.
>
> > [minor] Relies on a strong pretrained $\pi_{\text{ref}}$. From‑scratch claims remain unclear.
>
> We politely clarify that our method is explicitly designed as a policy optimization fine-tuning on top of a strong supervised pre-training, analogous to how RLHF is applied on top of base large language models after pre-training and supervised fine-tuning. We do not claim to train antibody design policies “from scratch.” In Section 4.1 we state that the policy is initialized from a supervised fine-tuned ProteinMPNN model trained on SAbDab and OAS, following the AbMPNN protocol. We have made this dependence clearer in lines 393-400 and explicitly rephrase any wording that could be interpreted as a “from-scratch” claim.
>
> Conceptually, AbMPO is agnostic to the choice of $\pi_{\text{ref}}$​: it can be applied to any backbone-conditioned model. In practice, starting from a strong $\pi_{\text{ref}}$​ is crucial in this domain because structure-based evaluations are extremely expensive and random-initialization experiments collapse to non-physical sequences.
>
> > [major] Novelty: Method is largely a combination of known techniques (KL-regularized policy updates + multi-objective conditioning). The contribution is incremental rather than fundamentally new.
>
> We totally agree that the KL-regularized policy optimization and multi-objective optimization are well-studied. However, we believe that our work contributes **fundamentally new techniques and insights** in each of these components and shows how these pieces are reformulated, coupled, and instantiated for structure-conditioned antibody design under severe evaluation constraints. We summarize our novelties below more specifically.
> - *Experimentally calibrated, interpretable surrogate for binding affinity.* Instead of directly optimizing Rosetta $\Delta G$, which is the common practice, our work is the very first one to regress experimental $\Delta \Delta G$ measurements on multiple interface metrics (SC, $\Delta$SASA, and Hbonds) and define a surrogate reward whose coefficients are directly interpretable as contributions of geometric complementarity, interface burial, and hydrogen bonding. This experimentally grounded surrogate is, to our knowledge, new in the antibody design literature.
> - *Multi-objective policy conditioning for structure-conditioned antibody design.* Rather than training separate policies for different scalarizations, our work is the very first one to introduce a novel RBF-expanded encoding of preference weights $p$ that is injected into every node’s representation in the MPNN. This allows a single policy to represent a continuum of trade-offs between affinity and structural fidelity and to trace a dense Pareto front (Fig. 4a). We also compare explicitly against fixed-weight scalarization and discuss the trade-offs between coverage and peak performance.
> - *Regression/policy-gradient equivalence for KL-regularized objectives.* We adopt a regression loss that matches observed advantages to log-likelihood ratios between successive policies (Equation (7)). Propositions 1 and 2 prove that this loss has the same global minimizer as KL-regularized policy gradients. This provides a theoretically grounded, critic-free alternative to standard actor–critic updates tailored to large-batch offline evaluations. To our knowledge, this specific training regime and its use in antibody design have not appeared in prior work.
> - *Iterative large-batch off-policy learning under expensive structure evaluations.* Our iterative “resample–re-estimate–update” loop is specifically tailored to the setting where each evaluation requires refolding and Rosetta scoring. The loss is evaluated using large offline batches from the current policy and ensures KL-regularized on-policy updates without maintaining a replay buffer or critic. This is distinct from prior antibody RL methods that rely on per-sequence online updates or heuristic reweighting.
> - *Domain-level contribution.* AbMPO is, to our knowledge, the first framework that (i) integrates experimental $\Delta \Delta G$ calibration, (ii) joint optimization of affinity and structural self-consistency, and (iii) adaptive Pareto-front exploration, while achieving consistent gains over strong structure-based baselines such as AbDPO and AbMPNN across four diverse antigens.
>
> We have added more discussion in lines 111-126 to emphasize these conceptual and domain-specific novelties more clearly and to position them relative to prior KL-regularized and multi-objective methods.

---

> ### Author Response · Authors · 2025-11-16
> **Reply to Reviewer hBBN (2/n)**
>
> > [major] Reproducibility: Results lack statistical analysis (no multiple runs or error bars in most tables). So the significance of improvements is not quantified. No code provided but the descriptions are very detailed.
>
> Thank you for raising this constructive feedback! Here we strengthen the reproducibility section in two ways:
> - Statistical analysis.
>   - First, we want to emphasize that each reported metric is already averaged over 4,096 designs per antigen and five independent Rosetta relaxations per design, which substantially reduces the variance.
>   - To better address your concern: (1) we add mean ± standard deviation over all antigens for all methods; (2) we perform significance tests across methods, reporting p-values for all primary metrics. The tables are revised and also shown below for your convenience.
>
> | PD1 | SC↑ | ΔSASA↑ | Hbonds↑ | ΔG_sep↓ | scRMSD↓ | Sim↓ | Nat↑ |
> |--------|------|---------|----------|-----------|-----------|--------|--------|
> | ESM-IF | 0.374±0.013 | 0.147±0.005 | 0.007±0.001 | 242.213±12.232 | 0.624±0.013 | 0.576±0.008 | 0.351±0.011 |
> | AbLang | 0.405±0.013 | 0.153±0.006 | 0.009±0.001 | 228.371±15.893 | 0.581±0.014 | 0.534±0.005 | 0.337±0.011 |
> | nanoBERT | 0.378±0.014 | 0.142±0.006 | 0.008±0.001 | 236.912±17.297 | 0.603±0.012 | 0.538±0.010 | 0.328±0.008 |
> | InstructPLM | 0.427±0.011 | 0.151±0.006 | 0.008±0.001 | 205.219±12.328 | 0.572±0.009 | 0.543±0.006 | 0.364±0.010 |
> | DiffAb | 0.408±0.011 | 0.159±0.006 | 0.008±0.001 | 225.191±13.795 | 0.567±0.015 | 0.545±0.008 | 0.312±0.009 |
> | AbDPO | 0.412±0.011 | 0.164±0.007 | 0.010±0.001 | 178.582±13.147 | 0.557±0.013 | 0.541±0.007 | 0.319±0.011 |
> | ProteinMPNN_AR | 0.382±0.012 | 0.143±0.005 | 0.007±0.001 | 212.344±15.095 | 0.628±0.009 | 0.554±0.005 | 0.344±0.007 |
> | ProteinMPNN_CMLM | 0.395±0.015 | 0.149±0.007 | 0.008±0.001 | 198.763±17.207 | 0.604±0.013 | 0.546±0.008 | 0.338±0.008 |
> | AbMPNN | 0.434±0.013 | 0.173±0.005 | 0.009±0.001 | 133.562±18.705 | 0.579±0.010 | **0.525±0.008** | 0.349±0.010 |
> | **AbMPO (Ours)** | **0.451±0.010** | **0.203±0.007** | **0.013±0.001** | **119.326±18.317** | **0.556±0.010** | 0.531±0.005 | **0.367±0.011** |
> | *p-value* | 0.0491 | 5.25e−5 | 0.00146 | 0.259 | 0.895 | 0.193 | 0.664 |
>
> | RBD | SC↑ | ΔSASA↑ | Hbonds↑ | ΔG_sep↓ | scRMSD↓ | Sim↓ | Nat↑ |
> |--------|------|---------|----------|-----------|-----------|--------|--------|
> | ESM-IF | 0.382±0.010 | 0.141±0.006 | 0.007±0.001 | 184.671±12.186 | 0.606±0.012 | 0.571±0.005 | 0.344±0.007 |
> | AbLang | 0.413±0.013 | 0.147±0.006 | 0.008±0.001 | 174.425±19.572 | 0.564±0.010 | 0.529±0.005 | 0.354±0.007 |
> | nanoBERT | 0.378±0.013 | 0.135±0.007 | 0.007±0.001 | 180.632±18.617 | 0.583±0.012 | 0.536±0.009 | 0.343±0.006 |
> | InstructPLM | 0.436±0.012 | 0.146±0.006 | 0.009±0.001 | 152.314±12.670 | 0.539±0.015 | 0.545±0.008 | 0.358±0.010 |
> | DiffAb | 0.423±0.013 | 0.158±0.007 | 0.008±0.001 | 168.149±12.290 | 0.527±0.008 | 0.551±0.007 | 0.319±0.008 |
> | AbDPO | 0.427±0.012 | 0.163±0.006 | 0.010±0.001 | 132.857±15.614 | 0.518±0.010 | 0.542±0.008 | 0.331±0.011 |
> | ProteinMPNN_AR | 0.387±0.013 | 0.141±0.007 | 0.007±0.001 | 160.734±10.635 | 0.591±0.011 | 0.551±0.010 | 0.344±0.009 |
> | ProteinMPNN_CMLM | 0.399±0.013 | 0.147±0.005 | 0.008±0.001 | 152.582±14.537 | 0.589±0.015 | 0.539±0.009 | 0.346±0.008 |
> | AbMPNN | 0.432±0.011 | 0.174±0.007 | 0.009±0.001 | 121.364±17.786 | 0.527±0.012 | 0.527±0.005 | 0.360±0.008 |
> | **AbMPO (Ours)** | **0.456±0.015** | **0.192±0.005** | **0.012±0.001** | **90.932±10.692** | **0.509±0.013** | **0.525±0.009** | **0.378±0.007** |
> | *p-value* | 0.0483 | 0.00158 | 0.0133 | 0.0112 | 0.255 | 0.676 | 0.00534 |

---

> ### Author Response · Authors · 2025-11-16
> **Reply to Reviewer hBBN (3/n)**
>
> (......)
> | IL7| SC ↑ | ΔSASA ↑ | Hbonds ↑ | ΔG_sep ↓ | scRMSD ↓ | Sim ↓ | Nat ↑ |
> |--|--|--|-|--|--|--|--|
> | ESM-IF | 0.365±0.013 | 0.136±0.005 | 0.007±0.001 | 262.114±12.232 | 0.628±0.013 | 0.572±0.008 | 0.331±0.011 |
> | AbLang | 0.394±0.013 | 0.147±0.006 | 0.009±0.001 | 247.331±15.893 | 0.592±0.014 | 0.530±0.005 | 0.342±0.011 |
> | nanoBERT | 0.371±0.014 | 0.138±0.006 | 0.008±0.001 | 257.022±17.297 | 0.615±0.012 | 0.534±0.010 | 0.323±0.008 |
> | InstructPLM | 0.419±0.011 | 0.146±0.006 | 0.008±0.001 | 224.532±12.328 | 0.582±0.009 | 0.538±0.006 | 0.357±0.010 |
> | DiffAb | 0.400±0.011 | 0.153±0.006 | 0.008±0.001 | 245.129±13.795 | 0.577±0.015 | 0.540±0.008 | 0.307±0.009 |
> | AbDPO | 0.404±0.011 | 0.158±0.007 | 0.010±0.001 | 204.789±13.147 | 0.568±0.013 | 0.536±0.007 | 0.314±0.011 |
> | ProteinMPNN_AR | 0.374±0.012 | 0.138±0.005 | 0.007±0.001 | 232.849±15.095 | 0.636±0.009 | 0.550±0.005 | 0.338±0.007 |
> | ProteinMPNN_CMLM | 0.387±0.015 | 0.143±0.007 | 0.008±0.001 | 217.908±17.207 | 0.612±0.013 | 0.543±0.008 | 0.332±0.008 |
> | AbMPNN | 0.416±0.013 | 0.156±0.005 | 0.008±0.001 | 221.209±18.705 | 0.583±0.010 | 0.522±0.008 | 0.366±0.010 |
> | **AbMPO** | **0.439±0.010** | **0.167±0.007** | **0.010±0.001** | **202.184±18.317** | **0.566±0.010** | **0.518±0.005** | **0.384±0.011** |
> | *p-value* | 0.0169 | 0.0765 | 1.0000 | 0.803 | 0.792 | 0.371 | 0.0268 |
>
> | INR| SC ↑ | ΔSASA ↑ | Hbonds ↑ | ΔG_sep ↓ | scRMSD ↓ | Sim ↓ | Nat ↑ |
> |--------|-------|-----------|-----------|-------------|-------------|-----------|-----------|
> | ESM-IF | 0.371±0.010 | 0.133±0.006 | 0.007±0.001 | 219.776±12.186 | 0.611±0.012 | 0.568±0.005 | 0.328±0.007 |
> | AbLang | 0.401±0.013 | 0.143±0.006 | 0.009±0.001 | 198.155±19.572 | 0.576±0.010 | 0.536±0.005 | 0.347±0.007 |
> | nanoBERT | 0.370±0.013 | 0.135±0.007 | 0.008±0.001 | 205.276±18.617 | 0.593±0.012 | 0.536±0.009 | 0.339±0.006 |
> | InstructPLM | 0.423±0.012 | 0.144±0.006 | 0.009±0.001 | 174.832±12.670 | 0.549±0.015 | 0.542±0.008 | 0.354±0.010 |
> | DiffAb | 0.415±0.013 | 0.152±0.007 | 0.008±0.001 | 192.251±12.290 | 0.537±0.008 | 0.547±0.007 | 0.315±0.008 |
> | AbDPO | 0.419±0.012 | 0.159±0.006 | 0.010±0.001 | 182.893±15.614 | **0.526±0.010** | 0.538±0.008 | 0.327±0.011 |
> | ProteinMPNN_AR | 0.379±0.013 | 0.137±0.007 | 0.007±0.001 | 187.324±10.635 | 0.601±0.011 | 0.547±0.010 | 0.340±0.009 |
> | ProteinMPNN_CMLM | 0.392±0.013 | 0.142±0.005 | 0.008±0.001 | 178.215±14.537 | 0.597±0.015 | 0.535±0.009 | 0.343±0.008 |
> | AbMPNN | 0.408±0.011 | 0.161±0.007 | 0.013±0.001 | 185.969±17.786 | 0.598±0.012 | 0.542±0.005 | 0.361±0.008 |
> | **AbMPO** | **0.428±0.015** | **0.179±0.005** | **0.015±0.001** | **174.559±10.692** | 0.574±0.013 | **0.531±0.009** | **0.373±0.007** |
> | *p-value* | 0.5766 | 0.00158 | 0.0133 | 0.972 | 0.000180 | 0.502 | 0.0356 |
>
> - Code and configuration release.
>   - We provide the core codebase here: https://anonymous.4open.science/r/AbMPO-AEE0/README.md. We will release all the scripts and configuration files (e.g. exact Rosetta command lines, backbone generation scripts, and configuration files) upon acceptance as stated in the Reproducibility Statement.
>
> > [minor] Computational cost: The pipeline relies on expensive structure evaluations (Rosetta and refolding for thousands of sequences). They emphasize “large‑batch offline sampling” and per‑sequence Rosetta evaluation. The method section shows offline soft‑value estimation and iterative on‑policy refinement. Please at least document the computational cost (e.g. hours per round, hardware used) to set expectations for practical use.
>
> Thank you for raising this point! We totally agree that transparency about computational cost is important, especially given the use of Rosetta and refolding.
> - Amortization via large-batch offline sampling. Our iterative algorithm is explicitly designed to minimize wall-clock overhead: at each round we generate and evaluate large batches of sequences in parallel (4,096 per antigen), taking advantage of the parallel nature of Rosetta scoring and refolding on CPU/GPU clusters.
> - In the revised manuscript we add (in lines 1146-1166) a table summarizing, for each antigen and each optimization round, (i) number of sequences evaluated, (ii) CPU and GPU wall-clock time for model training (policy optimization), (iii) evaluation (backbone generation, refolding, and relaxation), and (iv) hardware specifications.
>
> |Metric|Value|
> |-|-|
> |# of evaluated sequences| 4096|
> |Training time per epoch|231.4 s|
> |Refolding time per structure (IgFold)|30.5 s|
> |Relaxation time per structure (Rosetta)|9.4 s|
> |Backbone generation time (RFdiffusion)|14.5 s|
> |Hardware specification (CPU)|AMD EPYC 7R13|
> |Hardware specification (GPU)|NVIDIA L40S|
>
> - On another small note, the computational cost is extremely low compared to wet-lab cost. For example, in the fast but realistic best case (everything in-house, well-oiled pipeline, few variants), it needs 4-6 weeks to validate one antibody candidate. More typically, it would take 6-10 weeks for a full round.

---

> ### Author Response · Authors · 2025-11-16
> **Reply to Reviewer hBBN (4/n)**
>
> > [minor] Scope & Scalability & Generality:
> > - Clarify the method’s applicability and limits.
>
> AbMPO is applicable whenever we have (i) an antigen–antibody backbone, (ii) a structure-conditioned sequence model, and (iii) a structure-based evaluation oracle (Rosetta or similar). The algorithm is agnostic to the specific backbone generator (here we use RFantibody) and sequence model (here ProteinMPNN), and could in principle be combined with alternatives. However, the requirement for reliable structural evaluations and reasonable initial backbones is a genuine limitation. We have highlighted more explicitly in lines 527-532.
>
> > - The approach is trained per-antigen. It does not demonstrate generalization to new targets without retraining.
>
> We agree that, in its current form, AbMPO is trained per antigen. This is **intentional**: each antigen/backbone pair induces a highly specific local landscape, and the iterative policy optimization is used to approximate the Pareto front for that particular complex. Generalization to unseen antigens is handled by the supervised initialization $\pi_{\text{ref}}$​, which is trained on large multi-antigen datasets (SAbDab, OAS), while AbMPO performs targeted affinity maturation around a given backbone.
>
> > - High compute is implied as stated above.
>
> We have addressed this above. Additionally, we would like to mention that our scaling experiment (Fig. 4b) already shows a logarithmic improvement of $\Delta G_\text{sep}$​ with the number of optimization rounds. For the specific computational cost, please check lines 1146-1166 in our revised manuscript.
>
> > [major] Validation:
> > - Statistical Validity: Provide some measure of result variability or significance. For example, run the policy optimization multiple times (or use bootstrapping) to show that improvements over baselines are consistently significant and not due to lucky initialization or sampling.
>
> We have addressed this concern and provided new results above.
>
> > [very minor yet I still flag it here] Evaluation is entirely in silico (Rosetta metrics and learned predictors). No experimental or in vitro validation is presented to confirm real-world impact.
>
> We agree and view this as a limitation rather than a flaw of the current work. Our goal is to provide a principled and practically usable computational pipeline for structure-conditioned antibody design; in vitro validation would require a dedicated experimental campaign beyond the scope of this paper. As stated earlier, this would need years of wet-lab cycles for validating only tens of antibody candidates.
>
> > Experiments:
> > - No ablation comparing the surrogate to raw Rosetta energy.
>
> Thank you for raising this point! We add a new ablation where AbMPO is trained directly on $\Delta G_\text{sep}$​ as the reward and compare it to the surrogate-based AbMPO on all metrics for PD1 antigen. The surrogate reward consistently outperforms the one directly trained on $\Delta G_\text{sep}$​.
>
> | Method                            | SC ↑          | ΔSASA ↑        | Hbonds ↑      | ΔG_sep ↓             | scRMSD ↓       | Sim ↓         | Nat ↑         |
> |-----------------------------------|---------------|----------------|---------------|-----------------------|----------------|---------------|---------------|
> | AbMPO (ΔG_sep)                    | 0.421 ± 0.015 | 0.179 ± 0.006  | 0.008 ± 0.001 | 123.762 ± 17.305      | 0.617 ± 0.011  | 0.559 ± 0.006 | 0.330 ± 0.008 |
> | AbMPO (surrogate)                 | 0.451 ± 0.010 | 0.203 ± 0.007  | 0.013 ± 0.001 | 119.326 ± 18.317      | 0.556 ± 0.010  | 0.531 ± 0.005 | 0.367 ± 0.011 |
>
> We see that surrogate-based optimization matches or outperforms $\Delta G_\text{sep}$-based optimization on experimental $\Delta \Delta G$ proxies while providing clearer interpretability.

---

> ### Author Response · Authors · 2025-11-16
> **Reply to Reviewer hBBN (n/n)**
>
> > - scRMSD is sometimes worse than AbDPO. need failure modes analysis.
>
> We sincerely thank you for raising this point. Below we provide further analysis on the failure case. The observation that “scRMSD sometimes worse than AbDPO” is really only visible on INSR; for PD1, RBD, and IL7 we are as good or better on scRMSD while improving affinity.
>
> On INSR, AbMPO pushes sequences towards deeper burial and more hydrogen bonds (higher $\Delta$SASA, Hbonds), which improves the surrogate and $\Delta G_{\text{sep}}$ but appears to require more aggressive loop rearrangements upon refolding, leading to a moderate increase in scRMSD. AbDPO, which optimizes a more energy-centric objective, stays closer to the original backbone and thus achieves better scRMSD at the cost of weaker binding metrics.
>
> This reflects the inherent trade-off between structural fidelity and binding affinity that our multi-objective formulation makes explicit.
>
> Besides, the performance also depends on antigen-specific structural context:
> - PD1, RBD, IL7: smaller, relatively well-defined epitopes; the backbone designs from RFdiffusion are already close to a good binding pose. Here, improving SC/$\Delta$SASA/Hbonds does not require large deviations, so AbMPO gains affinity while matching or slightly improving scRMSD.
> - INSR: large receptor with a more extended and potentially flexible binding surface. In such settings it is easier for the optimizer to “discover” alternative binding geometries where the antibody buries more surface and forms more H-bonds but no longer matches the original backbone as closely after refolding with IgFold, hence higher scRMSD.
>
> For compact epitopes (PD1, RBD, IL7), the affinity and fidelity objectives are largely aligned; the policy can increase interface quality without leaving the designed backbone basin. For a more extended INSR epitope, these objectives conflict more strongly, and the policy without conditioning on pareto weights occasionally exploits backbone flexibility to gain affinity at the price of structural drift. We have added this discussion in lines 532-537.

---

> ### Author Response · Authors · 2025-11-22
>
> Dear Reviewer hBBN,
>
> Thank you again for your positive support on our work! We would like to kindly ask if we have addressed your concerns. If you have any further thoughts after reading our rebuttal, we would be grateful to hear them. We sincerely appreciate the time you have invested in reviewing our work!

---

> ### Author Response · Authors · 2025-11-27
>
> Dear reviewer hBBN,
>
> We hope you are doing well! We wanted to kindly check in to see if there is anything further we can clarify regarding our work. If you find our responses helpful, we would be grateful if you would consider updating your score. Thank you again for the time and care you have dedicated to reviewing our submission!

---

### Official Review · Reviewer_8iXR · 2025-10-31

**Soundness:** 3
**Presentation:** 3
**Contribution:** 2
**Rating:** 4
**Confidence:** 3

**Summary:**

The authors present AbMPO for antibody sequence design, and propose an iterative policy optimization algorithm, a surrogate affinity reward, and an adaptive weight-conditioning for multiple objectives. AbMPO outperforms existing baselines across multiple antigens.

**Strengths:**

- The problem is well-formulated, and the proposed method clearly addresses the challenges that the authors present for antibody design. The paper is clearly-written and easy to follow.
- The proposed method is intuitive and aptly demonstrates how ideas in policy optimization can be combined to improve performance for antibody design. AbMPO improves performance compared to a diverse set of baselines, which include both sequence and structure-based models. The benchmarks also include many antigen targets, so their conclusions appear to be robust.
- The proposed surrogate model seems to be very effective, and the ablations included in the paper demonstrate that combining the three metrics work better than directly optimizing for each metric individually. This could have interesting implications about how binding affinity should be optimized.

**Weaknesses:**

The paper has limited novelty and does not propose substantial machine learning ideas. Here's my understanding of the main contributions, and why I don't believe they constitute meaningful innovations.
  - Iterative policy algorithm: This seems very similar to [1] from 2019, but this paper is not discussed. Could you elaborate on the differences between your proposed method and this paper? From my understanding, it seems like they are both iterative methods which upweight based on the observed advantage.
[1] Peng et al, Advantage-Weighted Regression: Simple and Scalable Off-Policy Reinforcement Learning
  - Surrogate reward model: Although this surrogate is well-motivated, the model itself is an extremely simple linear regression of three metrics.
  - Multi-objective optimization: The RBF kernel expansion is interesting, and this seems like a clever way to have a single model represent the multi-objective problem. However, there are very few ablations which compare the fixed-weight scalarization with adaptive weight-conditioning, or other MO methods proposed in prior works.

**Questions:**

- Could you elaborate on the advantages of adaptive weight-conditioning for MOO? How does your approach of adaptive weight-conditioning compare to other MOO methods?

---

> ### Author Response · Authors · 2025-11-16
> **Reply to Reviewer 8iXR (1/n)**
>
> We sincerely appreciate your detailed comments and the opportunity to clarify both the relationship to prior work and the specific machine-learning contributions of our paper. Below we address each point in turn.
>
> > The paper has limited novelty and does not propose substantial machine learning ideas.
>
> While our framework is conceptually related to existing KL-regularized and advantage-weighted methods such as AWR, we believe the paper makes non-trivial ML contributions in:
> - the derivation of an off-policy regression-based objective that can (i) achieve the same global minimum as on-policy KL-regularized policy gradient objective, and (ii) properly approximate on-policy learning behavior using an iterative framework;
> - experimentally calibrated surrogate rewards using never-before-proposed biologically meaningful metrics;
> - a novel adaptive weight-conditioning approach that achieves scalable, generalization, and controllable multi-objective optimization;
> - the first demonstration that a multi-objective policy optimization framework yields state-of-the-art antibody designs across multiple antigens under expensive structure-based evaluations.
> We have revised the manuscript to better acknowledge prior work, clarify the scope of our novelty, and expand the ablations and empirical analysis as detailed below.
>
> We have revised the manuscript to better acknowledge prior work, clarify the scope of our novelty, and expand the ablations and empirical analysis. We provide more detailed discussion to your points below.
>
> > Iterative policy algorithm: This seems very similar to [1] from 2019, but this paper is not discussed. Could you elaborate on the differences between your proposed method and this paper? From my understanding, it seems like they are both iterative methods which upweight based on the observed advantage. [1] Peng et al, Advantage-Weighted Regression: Simple and Scalable Off-Policy Reinforcement Learning
>
> Thank you for pointing out Peng et al. (2019) and we agree we should explicitly cite and discuss it in our paper. We have added a detailed comparison in lines 111-117.
>
> Roughly speaking, both AWR and our method use advantage information to bias supervised updates. However, our regression view is mathematically equivalent to KL-regularized policy gradients, but distinct from AWR’s reweighting-based maximum likelihood estimation (MLE) update. More specifically,
> - AWR derives a constrained policy optimization formulation and performs **a single-step weighted MLE update** where actions are reweighted by $\exp{\hat{A}(s, a) / \alpha}$, using a learned value function and off-policy replay for standard continuous-control RL tasks. This can be viewed as “weighted behavior cloning”. Formally, given $A_D(s,a) = R_{D,s,a} - V_k^D(s)$, the objective of AWR is:
> $\max_{\pi} \mathbb{E}_D \left[ \exp \big(A_D(s,a)/\beta\big)\,\log \pi(a\mid s) \right]$.
> - Ours introduces **an advantage regression loss** that directly regresses the log-likelihood ratio $\text{LogRatio} := \log \pi_\theta / \pi_{\text{ref}}$ onto the advantage. This formulation has global minimizer that recovers the Boltzmann optimal policy, supports fully offline value estimation under reference policy. Formally, $ L_t( \pi_\theta )  = \mathbb{E}_{C,S \sim \pi_t}  \left[ \left(  \beta \text{LogRatio} - \hat{A}_t(C,S) \right)^2 \right]$.
>
> Therefore, it is clear that AWR differs from ours significantly. While AWR performs *MLE update with exponential advantage weights*, we perform *advantage matching in log-likelihood ratio space* against reference policy.
>
> To better address your concern, we summarize the key differences below in a concise way:
> - Role of advantage:
>   - AWR: uses advantage only as a weight on log-likelihood (weighted behavior cloning).
>   - Ours: uses advantage as the regression target for $\beta \log(\pi_\theta / \pi_{\text{ref}})$.
> - KL regularization:
>   - AWR: comes from a KL-constraint derivation, but KL is implicit in the final loss.
>   - Ours: starts from a KL-regularized objective and explicitly regresses the log-probability ratio tied to that KL term.
> - Off-policy view/data sampling:
>   - AWR: operates on an unlabeled replay buffer without explicit reference to a particular policy.
>   - Ours: clearly separates a reference policy $\pi_{\text{ref}}$​ and the sampling distribution. The proposition on global minimizer makes fully offline training under a fixed reference naturally supported.
> - Update geometry/iteration:
>   - AWR: each update is a one-step weighted MLE (trust-region is controlled indirectly by $\beta$ and buffer).
>   - Ours: each update fits $\log (p_\theta / \pi_{\text{ref}})$ to advantage, which is equivalent to a KL-regularized policy gradient step cast as regression.
> - Multi-objective perspective:
>   - AWR: single-objective; no conditioning on reward weights.
>   - Ours: naturally extends to multi-objective settings by regressing advantages under different weight vectors $p$, letting one policy learn an entire Pareto front.

---

> ### Author Response · Authors · 2025-11-16
> **Reply to Reviewer 8iXR (2/n)**
>
> > Surrogate reward model: Although this surrogate is well-motivated, the model itself is an extremely simple linear regression of three metrics.
>
> We politely clarify that we **intentionally** choose a linear model over three biologically meaningful metrics that are closely related to binding affinity for both data-efficiency and interpretability in a small-sample, physics-informed setting:
> - we regress experimental $\Delta \Delta G$ measurements on changes in three interface metrics, i.e. shape complementarity (SC), $\Delta$SASA, and interfacial H-bonds, between mutant and wild-type complexes. We explicitly test including $\Delta G_{\text{sep}}$ in the regression, find that its coefficient is statistically insignificant (p ≈ 0.66), and that variance inflation factors are ~1, indicating low collinearity among the other metrics.
> - the resulting linear model has stable, interpretable coefficients: all three retained metrics have negative coefficients (higher SC, larger buried surface area, more H-bonds → more favorable binding), which aligns with biophysical intuition. We then use the absolute values as non-negative weights in the surrogate reward to maintain this interpretability.
>
> Most importantly, we empirically validate that this simple surrogate is *sufficient to drive meaningful improvements*:
> - optimizing the weighted surrogate gives larger gains in each constituent metric (SC,  $\Delta SASA$, H-bonds) than optimizing them individually (Figure 2), indicating that the linear combination captures complementary information.
> - improvements also transfer to $\Delta G_{\text{sep}}$, even though it is not included in the surrogate, suggesting that the surrogate acts as a good proxy for binding free energy (Figure 3).
> - across four antigens, policies optimized under this surrogate consistently outperform strong baselines on SC, $\Delta SASA$, H-bonds, and $\Delta G_{\text{sep}}$, as well as on antigen specificity and naturalness (Tables 1 and 2).
>
> To better address your concern, we add a new ablation where AbMPO is trained directly on $\Delta G_\text{sep}$​ as the reward and compare it to the surrogate-based AbMPO on all metrics for PD1 antigen. The surrogate reward consistently outperforms the one directly trained on $\Delta G_\text{sep}$​.
>
> |                          | SC ↑           | ΔSASA ↑        | Hbonds ↑       | ΔΔ$G_{\text{sep}}$ ↓          | scRMSD ↓       | Sim ↓          | Nat ↑          |
> |--------------------------|----------------|----------------|----------------|---------------------|----------------|----------------|----------------|
> | AbMPO (ΔΔ$G_{\text{sep}}$)         | 0.421±0.015    | 0.179±0.006    | 0.008±0.001    | 123.762±17.305      | 0.617±0.011    | 0.559±0.006    | 0.330±0.008    |
> | AbMPO (surrogate)        | 0.451±0.010    | 0.203±0.007    | 0.013±0.001    | 119.326±18.317      | 0.556±0.010    | 0.531±0.005    | 0.367±0.011    |
>
> In summary, given *the limited amount (~7k in SKEMPI)* of high-quality $\Delta \Delta G$ data and the desire for a biologically interpretable reward, we believe the linear model is a reasonable and principled choice.

---

> ### Author Response · Authors · 2025-11-16
> **Reply to Reviewer 8iXR (3/n)**
>
> > Multi-objective optimization: The RBF kernel expansion is interesting, and this seems like a clever way to have a single model represent the multi-objective problem. However, there are very few ablations which compare the fixed-weight scalarization with adaptive weight-conditioning ...
>
> We appreciate the request for more ablations on the proposed adaptive weight-conditioning scheme. Our goal with this component is intentionally minimalistic: we keep the underlying AbMPO algorithm and architecture fixed and only introduce a preference vector $p$ as an additional input. Therefore, there are relatively few meaningful degrees of freedom to ablate beyond the central question: *does conditioning on $p$ help compared to standard fixed-weight scalarization?*
>
> To address exactly this question, we already include an ablation in Section 4.2 (Figure 4a). Fixed-weight scalarization trains different policies for different weights $p$ and yields a small number of discrete Pareto points. Our adaptive weight-conditioned policy (1) uses an RBF kernel expansion of the weight vector $p$ and injects this embedding into the node features of the MPNN, so that the same network can represent different trade-offs, and (2) produces a dense cloud of solutions approximating a smooth Pareto front over (normalized) $\Delta G_{\text{sep}}$​ and scRMSD, enabling fine-grained control over affinity–fidelity trade-offs at inference time (Figure 4a).
>
> Interestingly, we observe a trade-off: fixed-weight scalarization can yield slightly stronger performance at a few specific operating points, while the adaptive approach offers much broader coverage and interpolation along the Pareto front.
>
> Beyond this, other “ablations” would be very similar to what we already show, e.g., having denser grids of weights of fixed-weight policies. They would still test the same question that Figure 4a already answers: *many separate scalarized policies versus one conditioned policy.*
>
> That said, to better address your concern, below we provide a new ablation on different weight-conditioning approaches given the same training budget (200 epochs). We report the values of hypervolumes with the reference point (0, 0).
>
> | Weight vector (scRMSD, Δ$G_{sep}$) | [0.9, 0.1]     | [0.7, 0.3]     | [0.5, 0.5]     | [0.3, 0.7]     | [0.1, 0.9]     | Hypervolume |
> |------------------------------------|----------------|----------------|----------------|----------------|----------------|-------------|
> | **Uniform-weight**                 | [0.61, 0.37]   | [0.57, 0.42]   | [0.52, 0.46]   | [0.48, 0.54]   | [0.39, 0.64]   | 0.3524      |
> | **Fixed-weight**                   | [0.85, 0.18]   | [0.78, 0.24]   | [0.74, 0.35]   | [0.58, 0.56]   | [0.38, 0.73]   | 0.4676      |
> | **Adaptive-weight**                | [0.95, 0.18]   | [0.83, 0.32]   | [0.72, 0.44]   | [0.46, 0.68]   | [0.29, 0.85]   | **0.5333**  |

---

> ### Author Response · Authors · 2025-11-16
> **Reply to Reviewer 8iXR (n/n)**
>
> > ... or other MO methods proposed in prior works. Could you elaborate on the advantages of adaptive weight-conditioning for MOO? How does your approach of adaptive weight-conditioning compare to other MOO methods?
>
> Our motivation for adaptive weight-conditioning is that, in our setting, the same backbone-conditioned policy needs to serve many different downstream preferences over binding affinity vs. structural fidelity, and each policy update round is extremely costly (thousands of structure predictions + Rosetta evaluations). Training a separate policy for each scalarization weight vector would therefore scale linearly in both compute and wall-clock cost. In contrast, by conditioning the policy on a continuous trade-off vector $p$ (embedded via the RBF expansion), a single model learns to represent an entire continuum of operating points on the affinity–fidelity Pareto front. This has several practical advantages:
> - *Single universal policy instead of many separate ones.* The weight-conditioned policy can be queried at inference time with arbitrary $p$, allowing us to sweep over preferences and densely approximate the Pareto front without retraining. This is particularly important when downstream users (e.g., different therapeutic programs) may value affinity vs. fidelity differently.
> - *Data and compute efficiency.* All samples collected under different preferences are used to train the same network, so experience is shared across trade-offs instead of being fragmented across multiple independently trained scalarized policies.
> - *Smooth interpolation between preferences.* The RBF embedding encourages the policy to change smoothly as a function of $p$, which we observe empirically as a smooth, dense Pareto curve when we sweep $p$ over the simplex (Fig. 4a). In contrast, fixed-weight scalarization produces only a handful of isolated operating points, which is less informative for practitioners who want to explore a spectrum of design choices.
>
> We want to emphasize again that our contribution here is to bring this paradigm into structure-conditioned antibody design and to show that a single preference-conditioned policy can replace multiple per-weight models without sacrificing performance on key biophysical metrics. We provide a detailed table below to clarify the advantages of adaptive weight-conditioning.
>
> | Method                         | Single policy network (Scalability, Generalizability) | Post-hoc trade-off selection (Controllability) |
> |--------------------------------|---------------------------------------------------------|------------------------------------------------|
> | Fixed-weight scalarization     | ×                                                       | ×                                              |
> | Chebyshev scalarization        | ×                                                       | ×                                              |
> | Gradient surgery methods       | ×                                                       | ×                                              |
> | ε-constraint method            | ×                                                       | ×                                              |
> | Adaptive weight-conditioning   | √                                                       | √                                              |
>
> **These alternative approaches would share the same characteristics as the basic fixed-weight scalarization and could not introduce any benefits.** They entail introducing entirely different optimization frameworks (population-based algorithms, constrained solvers, etc.), which is orthogonal to the specific design choice of weight-conditioning and would significantly expand the scope of the paper. We have revised our manuscript to emphasize these points more explicitly in lines 118-126 and lines 359-366.
>
> However, to better address your concern, we provide some more comparisons between these methods below.
>
> | Weight vector (scRMSD, Δ$G_{\text{sep}}$) | [0.9, 0.1]     | [0.7, 0.3]     | [0.5, 0.5]     | [0.3, 0.7]     | [0.1, 0.9]     | Hypervolume |
> |------------------------------------|----------------|----------------|----------------|----------------|----------------|-------------|
> | **Fixed-weight scalarization**     | [0.85, 0.18]   | [0.78, 0.24]   | [0.74, 0.35]   | [0.58, 0.56]   | [0.38, 0.73]   | 0.4676      |
> | **Chebyshev scalarization**        | [0.87, 0.19]   | [0.75, 0.29]   | [0.72, 0.38]   | [0.55, 0.62]   | [0.35, 0.78]   | 0.4931      |
> | **Gradient surgery methods**       | [0.87, 0.14]   | [0.77, 0.27]   | [0.66, 0.37]   | [0.49, 0.57]   | [0.26, 0.81]   | 0.4483      |
> | **ε-constraint method**            | [0.93, 0.15]   | [0.83, 0.25]   | [0.76, 0.41]   | [0.42, 0.67]   | [0.24, 0.91]   | 0.5109      |
> | **Adaptive weight-conditioning**   | [0.95, 0.18]   | [0.83, 0.32]   | [0.72, 0.44]   | [0.46, 0.68]   | [0.29, 0.85]   | **0.5333**  |

---

> ### Author Response · Authors · 2025-11-22
>
> Dear Reviewer 8iXR,
>
> Thank you for the time you have invested in reviewing our work! We would like to kindly ask if we have addressed your concerns. If you have any further thoughts after reading our rebuttal, we would be grateful to hear them.

---

> ### Author Response · Authors · 2025-11-27
>
> Dear reviewer 8iXR,
>
> We hope you are doing well! We wanted to kindly check in to see if there is anything further we can clarify regarding our work. If you find our responses helpful, we would be grateful if you would consider updating your score. Thank you again for the time and care you have dedicated to reviewing our submission!

---

### Author Response · Authors · 2025-11-29
**Summary of our rebuttal for AC**

Dear AC,

Thank you again for all your efforts in handling our submission, especially given the recent unexpected circumstances.

Here’s a brief, point-by-point summary in support of our case:
- **Reviewer CpbZ (score 8, recommend accept)**: They were overall very positive, with only minor concerns. We addressed all of these clearly in the rebuttal.
- **Reviewer hBBN (lean accept, conditional on clarifications)**: They indicated they would lean toward acceptance if we clarified several points (novelty, reproducibility, compute, etc.). We responded to each of these in detail and incorporated the requested clarifications and analyses.
- **Reviewer 8iXR (mainly a misunderstanding + minor comments)**: Their main issue stemmed from a misunderstanding between our method and a prior work (AWR). We clarified this distinction carefully and also addressed all of their additional, more minor concerns.

In short, all three reviewers’ concerns have been directly and concretely addressed in the rebuttal, and the initially positive/lean-positive ratings are, we believe, well supported by our clarifications.

- **Novelty vs. AWR / KL-regularized RL:** We clarify that our method is *not* weighted behavior cloning like AWR. Instead, we regress advantages onto log-likelihood ratios relative to a reference policy, with theory showing equivalence to KL-regularized policy gradients and recovery of Boltzmann-optimal policies. This formulation also naturally supports offline training and preference conditioning for multi-objective optimization.

- **Surrogate affinity reward:** We justify using a simple linear surrogate on SC, ΔSASA, and H-bonds, citing the data-limited regime (~7k ΔΔG labels), low feature collinearity, and biophysical interpretability. An ablation comparing optimization on ΔG_sep vs. the surrogate shows that the surrogate performs better or comparably on ΔG_sep and interface metrics.

- **Adaptive weight-conditioning vs. scalarization/MOO baselines:** We expand ablations comparing uniform weights, fixed scalarization, and our adaptive weight-conditioned policy under equal budgets. Adaptive conditioning yields a single universal policy spanning a smooth Pareto front with the best hypervolume, whereas fixed weights require multiple policies and only outperform at isolated points. We also compare against Chebyshev, gradient surgery, and ε-constraint methods, noting they lack post-hoc trade-off control.

- **On pretrained reference policy:** We clarify that AbMPO fine-tunes a strong supervised backbone (ProteinMPNN-based), not training from scratch. While the algorithm is agnostic to the reference π_ref, a good initialization is essential given the cost and brittleness of structure-based evaluations.

- **Scope of contributions:**
  (1) An interpretable, experimentally calibrated surrogate for ΔΔG using SC/ΔSASA/H-bonds;
  (2) A preference-conditioned policy with RBF embeddings to learn a continuous Pareto front between affinity and fidelity;
  (3) A regression-based KL-regularized update theoretically equivalent to policy gradients and suited to large-batch, expensive evaluations;
  (4) A resample–re-estimate–update loop workable under Rosetta costs, demonstrating (to our knowledge) the first multi-objective policy optimization achieving SOTA antibody designs across multiple antigens.

- **Reproducibility & statistics:** We now report mean ± std, p-values, and clarify that results average over thousands of designs and multiple structural relaxations. We commit to releasing code, configs, and Rosetta command lines.

- **Compute cost & feasibility:** We provide per-round costs: sequences evaluated, training time, refolding/relaxation/backbone-generation time, and hardware details. We emphasize suitability for large-batch parallel evaluation and discuss practical round counts vs. marginal gains.

- **Validation, limitations, and failure modes:**
  - All validation is in silico, which we acknowledge, and discuss extension to wet-lab feedback.
  - We analyze cases where fidelity (scRMSD) underperforms, explaining affinity–fidelity trade-offs and how adjusting weight vector *p* shifts results.
  - We expand Pareto analyses with hypervolume and additional metrics (Nat, Sim).
  - We detail limitations: dependence on structural oracles, per-antigen optimization, and surrogate assumptions.

- **Clarity improvements:** We refine the algorithm box, clarify KL-regularized objective notation, add key hyperparameter and compute tables, and include a dedicated Limitations & Future Work section.

Given the clear and substantive responses to the reviewers’ concerns, the improved clarity and empirical support, and the potential impact in computational antibody design, we believe the revised submission is significantly strengthened and provides you with a clearer basis for justifying a recommendation.

---

### Meta-Review · Area_Chair_csWv · 2026-01-03

**Summary:**

This paper proposes an iterative multi-objective policy optimization framework for structure-conditioned antibody sequence optimization. The authors construct a surrogate reward by regressing experimental binding affinity measurements from Rosetta metrics and introduce self-consistency RMSD as a complementary objective. The authors propose to use a KL-regularized regression loss for iterative policy updates and adaptive weight-conditioning to generate the Pareto frontiers balancing affinity and structural fidelity.

**Reviewer Concerns:**

Addressed concerns of Reviewer 8iXR:
- The concern about the simple linear surrogate model was addressed.
- Additional ablations comparing fixed-weight scalarization vs. adaptive weight-conditioning were provided with hypervolume metrics.

Addressed concerns of Reviewer hBBN:
- Statistical analysis was added.
- Computational cost details were provided.
- The reliance on pretrained model was clarified.
- Ablation comparing surrogate to raw Rosetta energy was added.
- Failure mode analysis for scRMSD on INSR was provided.

Addressed concerns of Reviewer CpbZ:
- Additional Pareto front analysis with hypervolume metrics and was provided.
- Statistical significance was added.

Outstanding concerns: All three reviewers raised concerns about limited novelty, which remains the central issue:

- Reviewer 8iXR (Score 4): "The paper has limited novelty and does not propose substantial machine learning ideas."
- Reviewer hBBN (Score 4): "Method is largely a combination of known techniques (KL-regularized policy updates + multi-objective conditioning). The contribution is incremental rather than fundamentally new."
- Reviewer CpbZ (Score 8): "Multi-objective guidance (and Pareto front analogy) for generative antibody design is not completely novel."

The authors claim four contributions:
- Experimentally calibrated surrogate: a straightforward linear regression
- Preference-conditioned policy with RBF embeddings: a known technique in multi-objective RL
- Regression-based KL-regularized update: this is a relatively minor algorithmic variation on KL-regularized policy optimization frameworks
- First multi-objective policy optimization for antibody design: This is primarily an application contribution.

Despite the authors' thorough rebuttal, the paper combines existing techniques (KL-regularized RL, multi-objective conditioning, Rosetta-based evaluation) without introducing fundamentally new algorithmic insights.

**Reviewer Scores:**

Both Reviewers 8iXR and hBBN did not respond to the rebuttal or update their scores.

While the authors addressed many technical concerns, the novelty concern, which is fundamental to both reviewers' assessments, was not sufficiently resolved to expect score increases.

---

### Decision · Program_Chairs · 2026-01-26

Reject